# An Adaptive Quantum Circuit of Dempster's Rule of Combination for Uncertain Pattern Classification

**Fuyuan Xiao**[*]
School of Big Data and Software Engineering
Chongqing University
Chongqing, China 401331
xiaofuyuan@cqu.edu.cn

**Yu Zhou**
School of Big Data and Software Engineering
Chongqing University
Chongqing, China 401331

**Witold Pedrycz**
Department of Electrical and Computer Engineering
University of Alberta
Edmonton, AB T6G 2R3, Canada
Systems Research Institute, Polish Academy of Sciences
00-901 Warsaw, Poland,
National Information Processing Institute
00-608 Warsaw, Poland
wpedrycz@ualberta.ca

## Abstract

In pattern classification, efficient uncertainty reasoning plays a critical role, particularly in real-time applications involving noisy data, ambiguous class boundaries, or overlapping categories. Leveraging the advanced computational power of quantum computing, an Adaptive Quantum Circuit for Dempster's Rule of Combination (AQC-DRC) is proposed to address efficient classification under uncertain environments. The AQC-DRC is developed within the framework of quantum evidence theory (QET) and facilitates decision-making based on quantum basic probability and plausibility levels, which is a generalized Bayesian inference method. The AQC-DRC provides a deterministic computation of DRC, ensuring that quantum fusion outcomes in uncertain pattern classification are exactly aligned with those of the classical method, while simultaneously achieving exponential reductions in the computational complexity of evidence combination and significantly improving fusion efficiency. It is founded that the quantum basic probability amplitude function in QET, as a generalized quantum probability amplitude, can be naturally utilized to express the quantum amplitude encoding. In addition, the quantum basic probability in QET, as a generalized quantum probability, naturally forms a quantum basic probability distribution and can be used to represent quantum measurement outcomes for quantum basic probability level decision-making. Furthermore, the quantum plausibility function in QET also can be naturally used to express the quantum measurement outcomes for quantum plausibility level decision-making. These findings enrich the physical understanding of quantum amplitude encoding and quantum measurement outcomes, offering broad application prospects for representing and processing uncertain knowledge in pattern classification.

---

[*]Corresponding author: Fuyuan Xiao

39th Conference on Neural Information Processing Systems (NeurIPS 2025).

# 1 Introduction

Uncertainty and its dynamics is inherent in many real-world scenarios such as medical diagnosis, intelligent transportation, financial risk assessment, fault detection, and multi-sensor data fusion [1–6]. In the field of pattern classification, uncertainty reasoning is also essential, particularly when data are noisy, class boundaries are ambiguous, or there is overlap between categories. Properly modeling uncertainty in these situations contributes to enhanced robustness and reliability of classification systems [7–10]. To address uncertainty reasoning in such complex environments, numerous theories and methods have been proposed, including Dempster-Shafer evidence theory [11, 12], evidential reasoning [13, 14], D number theory [15], belief rule base [16], complex evidence theory [17], random permutation sets [18–22], and quantum evidence theory [23, 24], among others [25]. These approaches provide solid theoretical foundations and practical tools for modeling and managing uncertainty in various areas, such as evolutionary games [26, 27], network intrusion detection [28], classification [29, 30], social dilemma experiments [31–33], and reliability evaluation [34].

As an effective approach for uncertainty reasoning, DempsterShafer evidence theory (DSET) [11, 12] offers a powerful framework for representing and managing uncertainty through the basic probability assignment (BPA) function [35–37]. The Dempster's rule of combination (DRC) [11], a core component of DSET, possesses several desirable properties that make it particularly suitable for multisource information fusion [38–41], image segmentation [42]. time series [43], decision making [44, 45], engineering management [46], and others [47]. (1) Commutativity: ensures that the fusion result remains invariant regardless of the order in which evidence is combined; (2) Associativity: provides the system with flexible capabilities for structured and sequential fusion; and (3) Consistency: guarantees that, in the absence of new valid information, the outcome of the evidence combination remains unchanged. These advantages support flexible integration of multisource information, enable recursive and incremental computation, and facilitate the scalability of reasoning systems [48]. However, the computational complexity of DRC increases exponentially with the number of elements in the frame of discernment [49, 50].

The rapid development of quantum computing offers a new research perspective for addressing the computational complexity challenges in Dempster's rule of combination of DempsterShafer evidence theory [51–56]. Leveraging the principles of quantum parallelism and quantum entanglement, quantum computing provides the potential to significantly accelerate the processing of large-scale uncertain information with extensive applications [57–61]. In particular, it opens up new possibilities for overcoming the exponential computation complexity issues inherent in classical evidence reasoning frameworks based on DRC, thus providing an innovative approach to efficient information fusion and decision-making [62].

Leveraging the advanced computational power of quantum computing, an Adaptive Quantum Circuit for Dempster's Rule of Combination (AQC-DRC) is proposed to address efficient classification under uncertain environments. The AQC-DRC is developed within the framework of quantum evidence theory (QET) and facilitates decision-making based on quantum basic probability and plausibility levels, which is a generalized quantum Bayesian inference method. The AQC-DRC provides a deterministic computation of DRC, ensuring that quantum fusion outcomes in uncertain pattern classification are exactly aligned with those of the classical method, while simultaneously achieving exponential reductions in the computational complexity of evidence combination and significantly improving fusion efficiency.

In this study, it is founded that the quantum basic probability amplitude (QBPA) function in QET [23, 24], as a generalized quantum probability amplitude, can be naturally used to express the quantum amplitude encoding. In addition, the quantum basic probability (QBP) in QET, as a generalized quantum probability, naturally forms a quantum basic probability distribution (QBPD) and can be used to represent quantum measurement outcomes for basic probability level decision-making. Furthermore, the quantum plausibility (QPl) function in QET also can be naturally used to express the quantum measurement outcomes for quantum plausibility level decision-making. These findings open new perspectives and enrich the physical understanding of quantum amplitude encoding and quantum measurement outcomes, offering broad application prospects for representing and processing uncertain knowledge in areas such as pattern classification, recognition, and decision-making.

## 2 Preliminaries

### 2.1 DSET: DempsterShafer evidence theory [11, 12]

**Definition 1** (Frame of discernment)**.** *Let $\Omega$ be a frame of discernment (FOD), consisting of a set of mutually exclusive and collectively nonempty events:*

$$\Omega = \{h_1, h_2, \ldots, h_i, \ldots, h_n\}. \tag{1}$$

*Let $2^\Omega$ be the power set of $\Omega$, denoted as:*

$$2^\Omega = \{\emptyset, \{h_1\}, \{h_2\}, \ldots, \{h_n\}, \{h_1, h_2\}, \ldots, \{h_1, h_2, \ldots, h_i\}, \ldots, \Omega\}, \tag{2}$$

*where $\emptyset$ is an empty set [63–65].*

**Definition 2** (Hypothesis or proposition)**.** *$H_j$ is defined as a hypothesis or proposition when $H_j \subseteq \Omega$.*

**Definition 3** (Basic probability assignment)**.** *In FOD $\Omega$, a basic probability assignment (BPA) or basic belief assignment (BBA) $m$, also called a mass function, is defined as a mapping:*

$$m : \quad 2^\Omega \to [0, 1], \tag{3}$$

*satisfying*

$$m(\emptyset) = 0 \quad and \quad \sum_{H_j \subseteq \Omega} m(H_j) = 1. \tag{4}$$

**Definition 4** (Focal element)**.** *Let $m$ be a BPA. $\forall H_j \subseteq \Omega$, if $m(H_j) > 0$, $H_j$ is called a focal element in DSET.*

**Definition 5** (Plausibility function)**.** *Let $H_j$ and $H_t$ be two propositions such that $H_j, H_t \subseteq \Omega$. A plausibility function Pl, mapping from $2^\Omega$ to $[0, 1]$, is defined by*

$$Pl(H_j) = \sum_{H_t \cap H_j \neq \emptyset} m(H_t) = 1 - Bel(\bar{H}_j), \quad \bar{H}_j = \Omega - H_j, \tag{5}$$

*in which the belief function $Bel(\bar{H}_j) = \sum_{H_t \subseteq \bar{H}_j} m(H_t)$, measuring the strength of the evidence in favor of a proposition $\bar{H}_j$.*

**Definition 6** (Dempster's rule of combination)**.** *Let $m_1$ and $m_2$ be two independent BPAs in FOD $\Omega$ with propositions $H_t, H_h \subseteq \Omega$, respectively. Dempster's rule of combination (DRC), represented in the form $m_1 \oplus m_2$, is defined by*

$$m_1 \oplus m_2(H_j) = \begin{cases} \dfrac{1}{1-K} \displaystyle\sum_{H_t \cap H_h = H_j} m_1(H_t)m_2(H_h), & H_j \neq \emptyset, \\ 0, & H_j = \emptyset, \end{cases} \tag{6}$$

*with the conflict coefficient $K$ between $m_1$ and $m_2$:*

$$K = \sum_{H_t \cap H_h = \emptyset} m_1(H_t)m_2(H_h). \tag{7}$$

### 2.2 QET: Quantum evidence theory [23, 24]

**Definition 7** (Quantum frame of discernment)**.** *Let $\Phi$ be a quantum frame of discernment (QFOD), consisting of a set of mutually exclusive and collectively nonempty events, each of which is expressed as an orthonormal basis $\phi_g$ in a Hilbert space:*

$$\Phi = \{\phi_1, \ldots, \phi_g, \ldots, \phi_n\}. \tag{8}$$

*Let $2^\Phi$ be the power set of $\Phi$, denoted as:*

$$2^\Phi = \{\emptyset, \{\phi_1\}, \{\phi_2\}, \ldots, \{\phi_n\}, \{\phi_1\phi_2\}, \ldots, \{\phi_1\phi_2 \ldots \phi_g\}, \ldots, \Phi\}, \tag{9}$$

*which can be simply represented as ($\emptyset$ is an empty set):*

$$2^\Phi = \{\emptyset, \phi_1, \phi_2, \ldots, \phi_n, \phi_{12}, \ldots, \phi_{12\ldots g}, \ldots, \phi_{12\ldots n}\}. \tag{10}$$

**Definition 8** (Quantum hypothesis or proposition). $\psi_j$ *is defined as a quantum hypothesis or propo-sition when $\psi_j \subseteq \Phi$.*

**Definition 9** (Quantum basic probability amplitude function). *A quantum basic probability ampli-tude (QBPA) function $\mathbb{Q}_\mathbb{M}$ in QFOD $\Phi$, also referred to as a quantum mass function, is defined as a mapping:*

$$\mathbb{Q}_\mathbb{M} : 2^\Phi \to \mathbb{C}, \tag{11}$$

*satisfying*

$$\mathbb{Q}_\mathbb{M}(\emptyset) = 0 \quad and \quad \mathbb{Q}_\mathbb{M}(\psi_j) = \varphi(\psi_j)e^{i\theta(\psi_j)}, \quad \psi_j \subseteq \Phi,$$
$$\sum_{\psi_j \subseteq \Phi} |\mathbb{Q}_\mathbb{M}(\psi_j)|^2 = 1, \tag{12}$$

*in which $i = \sqrt{-1}$; $\varphi(\psi_j) \in [0, 1]$ represents the modulus of $\mathbb{Q}_\mathbb{M}(\psi_j)$; $\theta(\psi_j)$ denotes a phase term of $\mathbb{Q}_\mathbb{M}(\psi_j)$; and $|\mathbb{Q}_\mathbb{M}(\psi_j)|^2 = \varphi^2(\psi_j)$ denotes the modulus squared of $\mathbb{Q}_\mathbb{M}(\psi_j)$.*

Note that, QBPA is a generalized quantum probability amplitude. When the QBPA are only assigned to singleton states, the QBPA is referred to as quantum probability amplitude.

**Definition 10** (Quantum focal element). *Let $\mathbb{Q}_\mathbb{M}$ be a QBPA. $\forall \psi_j \subseteq \Phi$, if $|\mathbb{Q}_\mathbb{M}(\psi_j)|$ or $\varphi(\psi_j) > 0$, $\psi_j$ is called a quantum focal element in QET.*

**Definition 11** (Quantum basic probability function). *The quantum basic probability function of $\mathbb{Q}_\mathbb{M}$, also referred as a quantum basic probability distribution (QBPD), is defined as:*

$$\mathrm{M} : 2^\Phi \to [0, 1], \tag{13}$$

*and satisfies:*

$$\mathrm{M}(\emptyset) = 0 \quad and \quad \mathrm{M}(\psi_j) = |\mathbb{Q}_\mathbb{M}(\psi_j)|^2, \quad \psi_j \subseteq \Phi,$$
$$\sum_{\psi_j \subseteq \Phi} \mathrm{M}(\psi_j) = 1, \tag{14}$$

*where $|\mathbb{Q}_\mathbb{M}(\psi_j)|^2 = \mathbb{Q}_\mathbb{M}(\psi_j)\widehat{\mathbb{Q}}_\mathbb{M}(\psi_j) = \varphi^2(\psi_j) = x_j^2 + y_j^2$, in which $\widehat{\mathbb{Q}}_\mathbb{M}(\psi_j)$ is the complex conjugate of $\mathbb{Q}_\mathbb{M}(\psi_j)$, e.g., $\widehat{\mathbb{Q}}_\mathbb{M}(\psi_j) = x_j - y_j i$; and $\mathrm{M}(\psi_j)$ ($\psi_j \subseteq \Phi$) is called quantum basic probability (QBP), which represents the observed degree of belief or support to $\psi_j$.*

Note that, QBP is a generalized quantum probability. When the quantum basic probability are only assigned to singleton states, the QBP is referred to as quantum probability.

**Definition 12** (Quantum plausibility function). *Let $\mathbb{Q}_\mathbb{M}$ be a QBPA with proposition $\psi_j \subseteq \Phi$. A quantum plausibility (QPl) function in QET, mapping from $2^\Phi$ to $[0, 1]$, is defined by:*

$$QPl(\psi_j) = \sum_{\psi_p \cap \psi_j \neq \emptyset} \left| \mathbb{Q}_\mathbb{M}(\psi_p) \right|^2, \quad \psi_j \subseteq \Phi. \tag{15}$$

According to Eq. (14), Eq. (15) can also be represented as:

$$\mathrm{QPl}(\psi_j) = \sum_{\psi_p \cap \psi_j \neq \emptyset} \varphi^2(\psi_j) = \sum_{\psi_p \cap \psi_j \neq \emptyset} \mathrm{M}(\psi_p), \quad \psi_j \subseteq \Phi. \tag{16}$$

Therefore, when $\mathrm{M} = m$, Eq. (16) becomes:

$$\mathrm{QPl}(\psi_j) = \sum_{\psi_p \cap \psi_j \neq \emptyset} m(\psi_p), \quad \psi_j \subseteq \Phi, \tag{17}$$

which is consistent with the classical Pl in DSET.

# 3 AQC-DRC: Adaptive Quantum Circuit for Dempster's Rule of Combination

The AQC-DRC consists of the following three components: 1) quantum amplitude encoding for BPA, 2) construction of the adaptive quantum circuit for DRC, and 3) measurement in the adaptive quantum circuit for decision-making. The flowchart is displayed in Fig. 1.

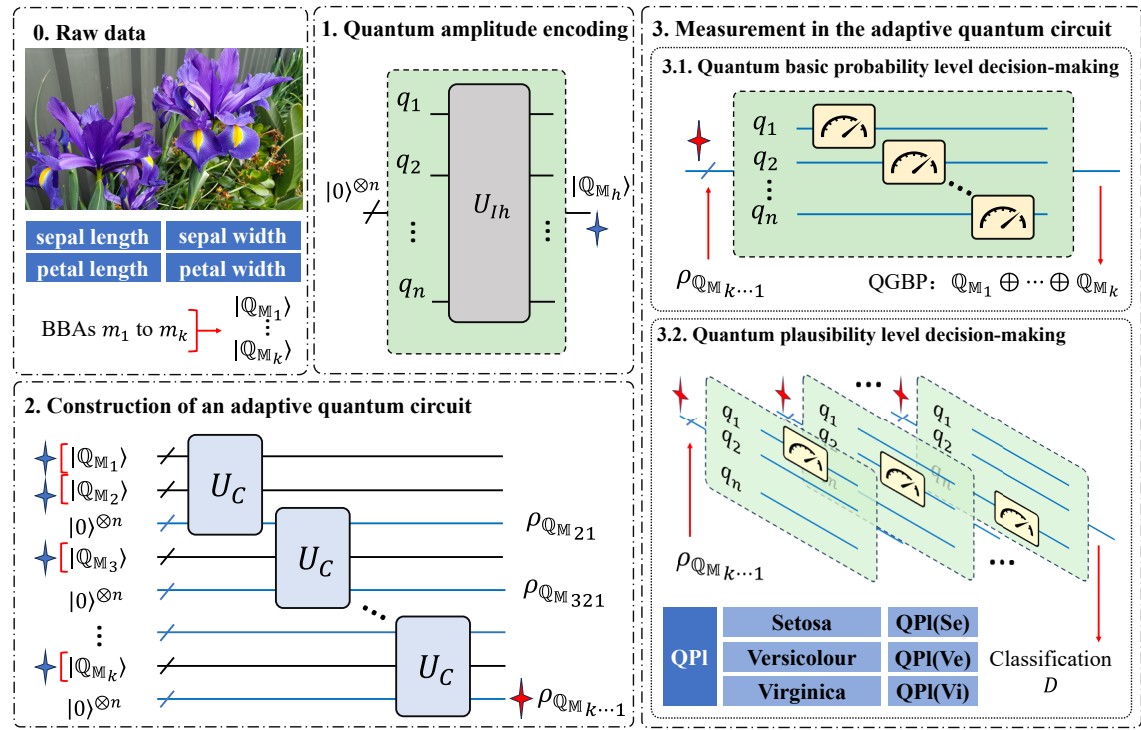

Figure 1: The flowchart of the adaptive quantum circuit for Dempster's rule of combination.

## 3.1 Quantum amplitude encoding for BPA

In this section, the QBPA in QET is expressed for quantum amplitude encoding mechanism. In this context, a BPA is encoded as a superposition over an $n$-qubit quantum state.

**Definition 13** (QBPA-based quantum amplitude encoding mechanism). *Let $\mathbb{Q}_{\mathbb{M}}$ be a QBPA on the QFOD $\Phi = \{\phi_1, \ldots, \phi_i, \ldots, \phi_n\}$ with quantum proposition $\psi_j \subseteq \Phi$. The QBPA-based quantum amplitude encoding mechanism, also referred to as the quantum superposition state of the QBPA, is defined as:*

$$|\mathbb{Q}_{\mathbb{M}}\rangle = \sum_{\psi_j \subseteq \Phi} \mathbb{Q}_{\mathbb{M}}(\psi_j)|\psi_j\rangle = \sum_{\psi_j \subseteq \Phi} \varphi(\psi_j)e^{i\theta(\psi_j)}|\psi_j\rangle,$$

$$\sum_{\psi_j \subseteq \Phi} |\mathbb{Q}_{\mathbb{M}}(\psi_j)|^2 = \sum_{\psi_j \subseteq \Phi} \varphi^2(\psi_j) = 1, \tag{18}$$

*where*

$$|\psi_j\rangle = \bigotimes_{i=1}^{n} |\delta_{ji}\rangle = |\delta_{jn}\rangle \cdots |\delta_{ji}\rangle \cdots |\delta_{j2}\rangle|\delta_{j1}\rangle = |\delta_{jn} \cdots \delta_{ji} \cdots \delta_{j2}\delta_{j1}\rangle, \tag{19}$$

*and*

$$\delta_{ji} = \begin{cases} 1, & \text{if } \phi_i \in \psi_j, \\ 0, & \text{if } \phi_i \notin \psi_j. \end{cases} \tag{20}$$

*When $\theta(\psi_j) = 0$, the QBPA expression for quantum amplitude encoding can be represented as:*

$$|\mathbb{Q}_{\mathbb{M}}\rangle = \sum_{\psi_j \subseteq \Phi} \varphi(\psi_j)|\psi_j\rangle, \quad \text{and} \quad \sum_{\psi_j \subseteq \Phi} \varphi^2(\psi_j) = 1. \tag{21}$$

**Definition 14** (Quantum amplitude encoding of BPA). *Let $m_h$ be a BPA on the FOD $\Phi = \{\phi_1, \ldots, \phi_i, \ldots, \phi_n\}$ with proposition $\psi_j \subseteq \Phi$. Considering QBPA-based quantum amplitude encoding mechanism, a BPA $m_h$ is encoded into the amplitudes of an n-qubit state as:*

$$|\mathbb{Q}_{\mathbb{M}_h}\rangle = U_E(m_h)|0\rangle^{\otimes n} = \sum_{\psi_j \subseteq \Phi} \mathbb{Q}_{\mathbb{M}_h}(\psi_j)|\psi_j\rangle = \sum_{\psi_j \subseteq \Phi} \varphi_h(\psi_j)e^{i\theta_h(\psi_j)}|\psi_j\rangle, \tag{22}$$

*satisfying*

$$\varphi_h(\psi_j)e^{i\theta_h(\psi_j)} = \sqrt{m_h(\psi_j)}e^{i0} = \sqrt{m_h(\psi_j)}, \quad and \quad \sum_{\psi_j \subseteq \Phi} \left| \sqrt{m_h(\psi_j)} \right|^2 = 1, \tag{23}$$

*where $|\psi_j\rangle$ is defined in Definition 13, and $U_E$ denotes a state preparation oracle or operator.*

Each BPA $m_h$ is encoded as a superposition over an $n$-qubit quantum state with regards to the QBPA-based quantum amplitude encoding mechanism. Therefore, a collection of $k$ BPAs $\{m_1, \ldots, m_h, \ldots, m_k\}$ is encoded to a set of $k$ distinct $n$-qubit quantum states $\{|\mathbb{Q}_{\mathbb{M}_1}\rangle, \ldots, |\mathbb{Q}_{\mathbb{M}_h}\rangle, \ldots, |\mathbb{Q}_{\mathbb{M}_k}\rangle\}$ with $|\mathbb{Q}_{\mathbb{M}_h}\rangle = U_E(m_h)|0\rangle^{\otimes n}$.

### 3.2  Construction of an adaptive quantum circuit for DRC

The encoded quantum states $\{|\mathbb{Q}_{\mathbb{M}_1}\rangle, \ldots, |\mathbb{Q}_{\mathbb{M}_h}\rangle, \ldots, |\mathbb{Q}_{\mathbb{M}_k}\rangle\}$ corresponding to the $k$ BPAs $\{m_1, \ldots, m_h, \ldots, m_k\}$ will be combined by a series of specific quantum operators, which is categorized as one types of $U_C$ designed through Toffoli gates.

To be specific, two encoded quantum states $|\mathbb{Q}_{\mathbb{M}_1}\rangle$ and $|\mathbb{Q}_{\mathbb{M}_2}\rangle$ are considered corresponding to the BPAs $m_1$ and $m_2$, the inputs of $U_C$ are $|\mathbb{Q}_{\mathbb{M}_1}\rangle$, $|\mathbb{Q}_{\mathbb{M}_2}\rangle$, and $|0\rangle^{\otimes n}$-qubit. After applying $U_C$ based on the Toffoli gate, the output is mapped to $|0\rangle^{\otimes n}$ qubits, which serve as auxiliary qubits for storing the processed information. Subsequently, a partial trace over the subsystem $\{\mathbb{Q}_{\mathbb{M}_2}, \mathbb{Q}_{\mathbb{M}_1}\}$ is performed using $\text{Tr}_{(\mathbb{Q}_{\mathbb{M}_2}\mathbb{Q}_{\mathbb{M}_1})}$ to obtain the reduced density matrix $\rho_{\mathbb{Q}_{\mathbb{M}_{21}}}$, which characterizes the state of the subsystem $\rho_{\mathbb{Q}_{\mathbb{M}_{21}}}$:

$$\begin{aligned}
\rho_{\mathbb{Q}_{\mathbb{M}_{21}}} &= \text{Tr}_{(\mathbb{Q}_{\mathbb{M}_2}\mathbb{Q}_{\mathbb{M}_1})} \left( U_C|0\rangle^{\otimes n}\langle 0|^{\otimes n} \otimes |\mathbb{Q}_{\mathbb{M}_2}\rangle\langle\mathbb{Q}_{\mathbb{M}_2}| \otimes |\mathbb{Q}_{\mathbb{M}_1}\rangle\langle\mathbb{Q}_{\mathbb{M}_1}|U_C^\dagger \right) \\
&= \text{Tr}_{(\mathbb{Q}_{\mathbb{M}_2}\mathbb{Q}_{\mathbb{M}_1})} \left( U_C|0\rangle^{\otimes n}\langle 0|^{\otimes n} \otimes \rho_{\mathbb{Q}_{\mathbb{M}_2}} \otimes \rho_{\mathbb{Q}_{\mathbb{M}_1}} U_C^\dagger \right) \\
&= \sum_{\psi_t \subseteq \Phi} \sum_{\cap\psi_j=\psi_t} \prod_{1\leq h\leq 2} \left| \varphi_h(\psi_j)e^{i\theta_h(\psi_j)} \right|^2 |\psi_t\rangle\langle\psi_t| \\
&= \sum_{\psi_t \subseteq \Phi} \sum_{\cap\psi_j=\psi_t} \prod_{1\leq h\leq 2} \left| \sqrt{m_h(\psi_j)} \right|^2 |\psi_t\rangle\langle\psi_t|,
\end{aligned} \tag{24}$$

where $|\psi_t\rangle$ is defined in Definition 13. It is concluded that for arbitrary two encoded quantum states corresponding to the BPAs, a $U_C$ operator of the Toffoli gate is implemented.

Hence, $h$ $(2 \leq h \leq k)$ encoded quantum states $\{|\mathbb{Q}_{\mathbb{M}_1}\rangle, \ldots, |\mathbb{Q}_{\mathbb{M}_h}\rangle\}$ are considered corresponding to the BPAs $\{m_1, \ldots, m_h\}$, $h-1$ $U_C$ operations based on Toffoli gates are recursively implemented in accordance with the DRC to fuse $h$ BPAs. Following the recursive application of the $(h-1)$-th $U_C$ Toffoli gate, we perform a partial trace over the subsystem $\{\mathbb{Q}_{\mathbb{M}_h}, \ldots, \mathbb{Q}_{\mathbb{M}_1}\}$ using $\text{Tr}_{(\mathbb{Q}_{\mathbb{M}_h}\ldots\mathbb{Q}_{\mathbb{M}_1})}$ to obtain the reduced density matrix $\rho_{\mathbb{Q}_{\mathbb{M}_{h\ldots 1}}}$, which characterizes the state of the corresponding subsystem:

$$\begin{aligned}
\rho_{\mathbb{Q}_{\mathbb{M}_{h\ldots 1}}} &= \text{Tr}_{(\mathbb{Q}_{\mathbb{M}_h}\mathbb{Q}_{\mathbb{M}_{(h-1)\ldots 1}})} \left( U_C|0\rangle^{\otimes n}\langle 0|^{\otimes n} \otimes \rho_{\mathbb{Q}_{\mathbb{M}_h}} \otimes \rho_{\mathbb{Q}_{\mathbb{M}_{(h-1)\ldots 1}}} U_C^\dagger \right) \\
&= \sum_{\psi_t \subseteq \Phi} \sum_{\cap\psi_j=\psi_t} \prod_{1\leq h\leq k} \left| \varphi_h(\psi_j)e^{i\theta_h(\psi_j)} \right|^2 |\psi_t\rangle\langle\psi_t| \\
&= \sum_{\psi_t \subseteq \Phi} \sum_{\cap\psi_j=\psi_t} \prod_{1\leq h\leq k} \left| \sqrt{m_h(\psi_j)} \right|^2 |\psi_t\rangle\langle\psi_t|,
\end{aligned} \tag{25}$$

where $|\psi_t\rangle$ is defined in Definition 13. When $h = k$, it indicates all the encoded quantum states corresponding to the BPAs are fused through the adaptive quantum circuit of DRC by $k - 1$ $U_C$ Toffoli gates. A total of $|0\rangle^{\otimes(2k-1)n}$ qubits are required to complete the DRC, where $|0\rangle^{\otimes kn}$ qubits are used for encoding the quantum states of the BPAs, and $|0\rangle^{\otimes(k-1)n}$ qubits are allocated as auxiliary qubits for storing the processed information.

### 3.3  Measurement in the adaptive quantum circuit for decision-making

We define two types of measurement operators in terms of the quantum basic probability level and the plausibility level decision-making for different application requirements.

### 3.3.1 Quantum measurement for quantum basic probability level decision-making

We first define a measurement operator for quantum basic probability level decision-making. After that, we define a QBP expression for quantum measurement outcomes. On this basis, we can measure the results from the adaptive quantum circuit of DRC.

**Definition 15** (Measurement operator for quantum basic probability level decision-making)**.** *The measurement operator $\mathcal{M}_{QBP}$ is defined for the quantum basic probability level decision-making as:*

$$\mathcal{M}_{QBP} = \{\mathcal{M}_{|\psi_t\rangle}|\psi_t \subseteq \Phi\}, \quad \mathcal{M}_{|\psi_t\rangle} = |\psi_t\rangle\langle\psi_t|, \tag{26}$$

*where $|\psi_t\rangle$ is defined in Definition 13.*

**Definition 16** (QBP expression for quantum measurement outcomes)**.** *Let $\rho_{\mathbb{Q}_{\mathbb{M}_{k\ldots 1}}}$ be a density operator with regards to the trace of the output of $U_C$. Let $\mathcal{M}_{QBP} = \{\mathcal{M}_{|\psi_t\rangle} = |\psi_t\rangle\langle\psi_t||\psi_t \subseteq \Phi\}$ be a set of measurement operators. The quantum basic probability (QBP) expression for quantum measurement outcomes is defined as:*

$$\mathrm{M}(\psi_t) = \frac{\mathrm{Tr}\left(\mathcal{M}_{|\psi_t\rangle}^{\dagger}\mathcal{M}_{|\psi_t\rangle} \cdot \rho_{\mathbb{Q}_{\mathbb{M}_{k\ldots 1}}}\right)}{\sum\limits_{\substack{\psi_v \subseteq \Phi \\ \psi_v \neq \emptyset}} \mathrm{Tr}\left(\mathcal{M}_{|\psi_v\rangle}^{\dagger}\mathcal{M}_{|\psi_v\rangle} \cdot \rho_{\mathbb{Q}_{\mathbb{M}_{k\ldots 1}}}\right)} = \frac{\sum\limits_{\cap\psi_j=\psi_t} \prod\limits_{1\le h\le k} \left|\varphi_h(\psi_j)e^{i\theta_h(\psi_j)}\right|^2}{\sum\limits_{\substack{\psi_v \subseteq \Phi \\ \psi_v \neq \emptyset}} \sum\limits_{\cap\psi_j=\psi_v} \prod\limits_{1\le h\le k} \left|\varphi_h(\psi_j)e^{i\theta_h(\psi_j)}\right|^2}, \tag{27}$$

*which forms a quantum basic probability distribution (QBPD)* $\mathrm{M}$.

After implementing the measurement operator $\mathcal{M}_{QBP}$, for $\psi_t \subseteq \Phi, \psi_t \neq \emptyset$, because $\left|\varphi_h(\psi_j)e^{i\theta_h(\psi_j)}\right|^2 = \left|\sqrt{m_h(\psi_j)}\right|^2$ and $m(\psi_t) = \mathrm{M}(\psi_t)$, the combined BPA can be generated:

$$m(\psi_t) = \mathrm{M}(\psi_t) = \frac{\sum\limits_{\cap\psi_j=\psi_t} \prod\limits_{1\le h\le k} \left|\sqrt{m_h(\psi_j)}\right|^2}{\sum\limits_{\substack{\psi_v \subseteq \Phi \\ \psi_v \neq \emptyset}} \sum\limits_{\cap\psi_j=\psi_v} \prod\limits_{1\le h\le k} \left|\sqrt{m_h(\psi_j)}\right|^2}. \tag{28}$$

For $\psi_t = \emptyset$, we have

$$\begin{aligned} K &= \mathrm{Tr}\left(\mathcal{M}_{|\emptyset\rangle}^{\dagger}\mathcal{M}_{|\emptyset\rangle} \cdot \rho_{\mathbb{Q}_{\mathbb{M}_{k\ldots 1}}}\right) \\ &= \sum\limits_{\cap\psi_j=\emptyset} \prod\limits_{1\le h\le k} \left|\varphi_h(\psi_j)e^{i\theta_h(\psi_j)}\right|^2 = \sum\limits_{\cap\psi_j=\emptyset} \prod\limits_{1\le h\le k} \left|\sqrt{m_h(\psi_j)}\right|^2. \end{aligned} \tag{29}$$

Then, for $\psi_t \subseteq \Phi, \psi_t \neq \emptyset$, we also have

$$m(\psi_t) = \frac{\mathrm{Tr}\left(\mathcal{M}_{|\psi_t\rangle}^{\dagger}\mathcal{M}_{|\psi_t\rangle} \cdot \rho_{\mathbb{Q}_{\mathbb{M}_{k\ldots 1}}}\right)}{1 - \mathrm{Tr}\left(\mathcal{M}_{|\emptyset\rangle}^{\dagger}\mathcal{M}_{|\emptyset\rangle} \cdot \rho_{\mathbb{Q}_{\mathbb{M}_{k\ldots 1}}}\right)} = \frac{\sum\limits_{\cap\psi_j=\psi_t} \prod\limits_{1\le h\le k} \left|\sqrt{m_h(\psi_j)}\right|^2}{1 - K}. \tag{30}$$

When implementing AQC-DRC based on the quantum measurement for quantum basic probability level decision-making, denoted as AQC-DRC$_{\text{QBP}}$, for $\||\psi_t\rangle\| = 1$, a decision can be made as follow:

$$w = \arg\max_t\{\mathrm{M}(\psi_t)\} = \arg\max_t\{m(\psi_t)\}, \quad \text{and} \quad D = \psi_w. \tag{31}$$

### 3.3.2 Quantum measurement for quantum plausibility level decision-making

Starting from the density matrix $\rho_{\mathbb{Q}_{\mathbb{M}_{h\ldots 1}}}$, we perform a partial trace over the auxiliary $(n-1)$ qubits $\{q_n \ldots q_{w+1}q_{w-1} \ldots q_1\}$ using $\mathrm{Tr}_{(q_n\ldots q_{w+1}q_{w-1}\ldots q_1)}$. This yields the reduced density matrix

$\rho^w_{\mathbb{Q}_{\mathbb{M}_{h\ldots1}}}$, which describes the state associated with the $w$-th qubit:

$$\rho^w_{\mathbb{Q}_{\mathbb{M}_{h\ldots1}}} = \mathrm{Tr}_{(q_n\ldots q_{w+1}q_{w-1}\ldots q_1)}\left(\rho_{\mathbb{Q}_{\mathbb{M}_{h\ldots1}}}\right)$$

$$= \sum_{\phi_w\in\psi_t}\sum_{\cap\psi_j=\psi_t}\prod_{1\leq h\leq k}\left|\varphi_h(\psi_j)e^{i\theta_h(\psi_j)}\right|^2 |1\rangle\langle1| + \left(1 - \sum_{\phi_w\in\psi_t}\sum_{\cap\psi_j=\psi_t}\prod_{1\leq h\leq k}\left|\varphi_h(\psi_j)e^{i\theta_h(\psi_j)}\right|^2\right)|0\rangle\langle0|$$

$$= \sum_{\phi_w\in\psi_t}\sum_{\cap\psi_j=\psi_t}\prod_{1\leq h\leq k}\left|\sqrt{m_h(\psi_j)}\right|^2 |1\rangle\langle1| + \left(1 - \sum_{\phi_w\in\psi_t}\sum_{\cap\psi_j=\psi_t}\prod_{1\leq h\leq k}\left|\sqrt{m_h(\psi_j)}\right|^2\right)|0\rangle\langle0|.$$

$$(32)$$

**Definition 17** (Measurement operator for quantum plausibility level decision-making). *The measurement operator $\mathcal{M}_{QPl}$ is defined for quantum plausibility level decision-making as:*

$$\mathcal{M}_{QPl} = \{\mathcal{M}_{|u\rangle}|u\in\{0,1\}\}, \quad \mathcal{M}_{|u\rangle} = |u\rangle\langle u|. \tag{33}$$

**Definition 18** (QPl expression for quantum measurement outcomes). *Let $\rho^w_{\mathbb{Q}_{\mathbb{M}_{k\ldots1}}}$ be the density operator of the $w$-th qubit in terms of the output of $U^C$. Let $\mathcal{M}_{QPl} = \{\mathcal{M}_{|u\rangle} = |u\rangle\langle u||u\in\{0,1\}\}$ and $\mathcal{M}_{QBP} = \{\mathcal{M}_{|\psi_t\rangle} = |\psi_t\rangle\langle\psi_t||\psi_t\subseteq\Phi\}$ be a set of measurement operators. The quantum plausibility (QPl) expression for quantum measurement outcomes is defined as:*

$$\mathrm{QPl}(\psi_w) = \frac{\mathrm{Tr}\left(\mathcal{M}^\dagger_{|1\rangle}\mathcal{M}_{|1\rangle}\cdot\rho^w_{\mathbb{Q}_{\mathbb{M}_{k\ldots1}}}\right)}{1 - \mathrm{Tr}\left(\mathcal{M}^\dagger_{|\emptyset\rangle}\mathcal{M}_{|\emptyset\rangle}\cdot\rho^w_{\mathbb{Q}_{\mathbb{M}_{k\ldots1}}}\right)} = \frac{\sum_{\phi_w\in\psi_t}\sum_{\cap\psi_j=\psi_t}\prod_{1\leq h\leq k}\left|\varphi_h(\psi_j)e^{i\theta_h(\psi_j)}\right|^2}{1 - K}. \tag{34}$$

After implementing the measurement operator $\mathcal{M}_{QPl}$, because $\left|\varphi_h(\psi_j)e^{i\theta_h(\psi_j)}\right|^2 = \left|\sqrt{m_h(\psi_j)}\right|^2$ and $\mathrm{Pl}(\psi_w) = \mathrm{QPl}(\phi_w)$ for each element $\phi_w$ $(1\leq w\leq n)$ in FOD, the Pl for $\phi_w$ can be generated:

$$\mathrm{Pl}(\phi_w) = \mathrm{QPl}(\phi_w) = \frac{\sum_{\phi_w\in\psi_t}\sum_{\cap\psi_j=\psi_t}\prod_{1\leq h\leq k}\left|\sqrt{m_h(\psi_j)}\right|^2}{1 - K}. \tag{35}$$

Since $1/(1 - K)$ is a constant normalization factor, it does not affect the decision-making outcome, and we can simply Eq. (34) for decision-making to improve computational efficiency witout loss of generality:

$$\widehat{\mathrm{QPl}}(\psi_w) = \mathrm{Tr}\left(\mathcal{M}^\dagger_{|1\rangle}\mathcal{M}_{|1\rangle}\cdot\rho^w_{\mathbb{Q}_{\mathbb{M}_{k\ldots1}}}\right). \tag{36}$$

When implementing AQC-DRC based on the quantum plausibility level decision-making, denoted as AQC-DRC$_{\mathrm{QPl}}$, a decision can be made as follow:

$$w = \arg\max_w\{\widehat{\mathrm{QPl}}(\phi_w)\} = \arg\max_w\{\mathrm{Pl}(\phi_w)\}, \quad \text{and} \quad D = \phi_w. \tag{37}$$

Therefore, based on the definitions and rigorous mathematical derivations in Sections 3.1-3.3, we can observe that the proposed AQC-DRC$_{\mathrm{QBP}}$ yields results consistent with DRC at the quantum basic probability level, while AQC-DRC$_{\mathrm{QPl}}$ aligns with DRC+QPl at the quantum plausibility level. These results demonstrate deterministic computation and guarantee that no information is lost.

## 4 Application in pattern classification

In the experimental setup, three distinct datasets namely Iris (Ir), Abalone (Ab), and Knowledge (Kn) are chosen from the UCI repository (https://archive.ics.uci.edu/). For the Iris dataset, all attributes and classes are selected. Whereas, due to the limitation of available qubits, for the Abalone dataset, we randomly select five attributes (Length, Diameter, Height, Whole_weight, and Viscera_weight) and two classes (Rings = 5 and Rings = 15); for the Knowledge dataset, all classes are selected, but three attributes (STR, LPR, and PEG) are randomly select. The details of each dataset, including the selected attributes and classes, are summarized in Table 1. The approach from [66] is adopted to generate the BPA for each attribute. The individual BPAs of each dataset are

subsequently aggregated to perform classification using different methods. To evaluate the effectiveness of the proposed methods, we compare the classical DRC [11], DRC+QPl, QC-DRC$_{QBP}$ [62], QC-DRC$_{QBP}$+QPl (the QC-DRC$_{QBP}$ is used firstly to obtain the QBP, and then to combine the QBP into the QPl using the classical process), and the proposed AQC-DRC$_{QBP}$, AQC-DRC$_{QBP}$+QPl (the AQC-DRC$_{QBP}$ is used firstly to obtain the QBP, and then to combine the QBP into the QPl using the classical process) and AQC-DRC$_{QPl}$ based on the average accuracy (Acc) obtained from $\kappa$-fold cross-validation ($\kappa = 5$). To analyze the robustness of different methods, we further evaluate the average accuracy performance of each method under varying numbers of measurement shots ($s$).

<table>
<tr><td colspan="4">Table 1: Dataset information</td></tr>
<tr><td></td><td>Ir</td><td>Ab</td><td>Kn</td></tr>
<tr><td># Class</td><td>3</td><td>2</td><td>4</td></tr>
<tr><td># Attribute</td><td>4</td><td>5</td><td>3</td></tr>
<tr><td># Instance</td><td>150</td><td>218</td><td>403</td></tr>
</table>

Table 2: Computational complexity analysis

| Method | Complexity | Qubits |
|---|---|---|
| DRC | $O(kN2^{2n})$ | / |
| DRC + QPl | $O(kN2^{2n})$ | / |
| QC-DRC$_{QBP}$ | $O(kn + kN)$ | $3n$ |
| QC-DRC$_{QBP}$ + QPl | $O(kn + kN + nN)$ | $3n$ |
| AQC-DRC$_{QBP}$ | $O(kn + N)$ | $(2k-1)n$ |
| AQC-DRC$_{QBP}$ + QPl | $O(kn + nN)$ | $(2k-1)n$ |
| AQC-DRC$_{QPl}$ | $O(kn^2)$ | $(2k-1)n$ |

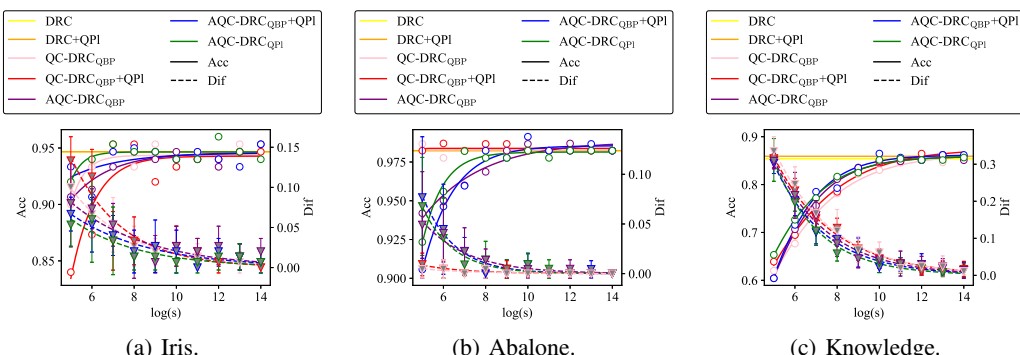

(a) Iris.  (b) Abalone.  (c) Knowledge.

Figure 2: Comparison of the classical and quantum methods in terms of Acc and Dif metrics.

The experimental results are displayed in Fig. 2 and Tables A1-A6. It is observed that as the number of shots $s$ increases from $2^5$ to $2^6$, $2^7$, $2^8$, $2^9$, $2^{10}$, $2^{11}$, $2^{12}$, $2^{13}$, and finally $2^{14}$, for Iris dataset, the average accuracies of QC-DRC$_{QBP}$ change as follows: 0.9, 0.9333, 0.9467, 0.9333, 0.9533, 0.9333, 0.9467, 0.9333, 0.9533, and 0.9467; the average accuracies of QC-DRC$_{QBP}$+QPl change as follows: 0.9333, 0.9067, 0.9467, 0.95, 0.9467, 0.94, 0.9467, 0.94, 0.9333, and 0.9533; the average accuracies of AQC-DRC$_{QBP}$ change as follows: 0.9067, 0.9133, 0.9333, 0.9467, 0.94, 0.9467, 0.94, 0.9333, 0.9467, and 0.9533; the average accuracies of AQC-DRC$_{QBP}$+QPl change as follows: 0.92, 0.94, 0.9533, 0.9467, 0.9467, 0.94, 0.94, 0.96, 0.9467, and 0.94; the average accuracies of AQC-DRC$_{QPl}$ change as follows: 0.84, 0.8733, 0.9533, 0.9533, 0.92, 0.9333, 0.94, 0.94, 0.9467, and 0.9467. While the classical DRC and DRC+QPl maintain an average accuracy of 0.9467, the accuracies of QC-DRC$_{QBP}$, QC-DRC$_{QBP}$+QPl, the proposed AQC-DRC$_{QBP}$, AQC-DRC$_{QBP}$+QPl, and AQC-DRC$_{QPl}$ gradually approach those of the classical models. Similarly, on the Abalone dataset, classical DRC and DRC+QPl achieve accuracies of 0.9823; on the Knowledge dataset, they reach 0.8539 and 0.8591, respectively. The corresponding quantum methods, including QC-DRC$_{QBP}$, QC-DRC$_{QBP}$+QPl, AQC-DRC$_{QBP}$+QPl, and AQC-DRC$_{QPl}$, also demonstrate a consistent trend of converging toward the classical models' performance.

On the other hand, we define an evaluation metric, Dif, to quantify the difference between the classification results of the classical DRC, and those produced by the QC-DRC$_{QBP}$, QC-DRC$_{QBP}$+QPl, the proposed AQC-DRC$_{QBP}$, AQC-DRC$_{QBP}$+QPl, and AQC-DRC$_{QPl}$. The metric Dif$_s$ under different numbers of shots $s$ for $k$-fold cross-validation is defined as:

$$\text{Dif}_s \left( \frac{\text{Method}}{\text{DRC}} \right) = \frac{1}{5} \sum_{\kappa=1}^{5} \frac{|\text{Mc}_s^\kappa(\text{Method}) \cup \text{Mc}^\kappa(\text{DRC}) - \text{Mc}_s^\kappa(\text{Method}) \cap \text{Mc}^\kappa(\text{DRC})|}{|\text{Tc}^\kappa|}, \quad (38)$$

where $\text{Mc}_s^\kappa(\text{Method})$ denotes the misclassified instances by other methods, including QC-DRC$_{\text{QBP}}$, QC-DRC$_{\text{QBP}}$+QPl, the proposed AQC-DRC$_{\text{QBP}}$, AQC-DRC$_{\text{QBP}}$+QPl, and AQC-DRC$_{\text{QPl}}$; $\text{Mc}^\kappa(\text{DRC})$ represents the misclassified instances by the classical DRC; $\text{Tc}^\kappa$ denotes the test instances in the $\kappa$-th fold of cross-validation; and $|\cdot|$ represents the number of instances. StD denotes the standard deviation of $\text{Dif}_s$ with respect to $\kappa$-fold cross-validation. The experimental results are displayed in Fig. 2 and Tables A7-A12. It is observed that as the number of shots $s$ increases from $2^5$ to $2^{14}$, the Dif values of the QC-DRC$_{\text{QBP}}$, QC-DRC$_{\text{QBP}}$+QPl, the proposed AQC-DRC$_{\text{QBP}}$, AQC-DRC$_{\text{QBP}}$+QPl, and AQC-DRC$_{\text{QPl}}$ relative to the classical DRC gradually converge to zero in terms of three distinct datasets.

## 5 Computational complexity analysis

Assume that there are $n$ elements in the frame of discernment (FOD) and $k$ pieces of evidence, with a total of $N$ focal elements. As shown in Table 2, the time complexities of the classical DRC and DRC+QPl are $O(kN2^{2n})$. In contrast, with sufficient auxiliary qubits, the time complexities of QC-DRC$_{\text{QBP}}$, QC-DRC$_{\text{QBP}}$+QPl, AQC-DRC$_{\text{QBP}}$, AQC-DRC$_{\text{QBP}}$+QPl and AQC-DRC$_{\text{QPl}}$, in terms of both the circuit depth and the normalization process, are $O(kn + kN)$, $O(kn + kN + nN)$, $O(kn + N)$, $O(kn + nN)$ and $O(kn^2)$, respectively. Comparative analysis reveals that QC-DRC$_{\text{QBP}}$, QC-DRC$_{\text{QBP}}$+QPl, AQC-DRC$_{\text{QBP}}$, AQC-DRC$_{\text{QBP}}$+QPl and AQC-DRC$_{\text{QPl}}$ all achieve exponential reductions in time complexity relative to the classical DRC and DRC+QPl. Notably, AQC-DRC$_{\text{QBP}}$ improves upon QC-DRC$_{\text{QBP}}$ by reducing the time complexity by $(k-1)N$; AQC-DRC$_{\text{QBP}}$+QPl improves upon QC-DRC$_{\text{QBP}}$+QPl by reducing the time complexity by $kN$. Furthermore, since $1 \leq N \leq 2^n$, when $N = 2^n$, AQC-DRC$_{\text{QPl}}$ achieves an exponential improvement over QC-DRC$_{\text{QBP}}$, QC-DRC$_{\text{QBP}}$+QPl, AQC-DRC$_{\text{QBP}}$ and AQC-DRC$_{\text{QBP}}$+QPl. Regarding space complexity, QC-DRC$_{\text{QBP}}$, QC-DRC$_{\text{QBP}}$+QPl require $3n$ qubits. In comparison, AQC-DRC$_{\text{QBP}}$, AQC-DRC$_{\text{QBP}}$+QPl and AQC-DRC$_{\text{QPl}}$ require $(2k-1)n$ qubits, growing linearly with both the number of elements $n$ in the FOD and the number of evidence sources $k$. The results indicate that the proposed AQC-DRC$_{\text{QBP}}$, AQC-DRC$_{\text{QBP}}$+QPl and AQC-DRC$_{\text{QPl}}$ exhibit superior time complexity compared to existing approaches. Specifically, AQC-DRC$_{\text{QPl}}$ demonstrates optimal performance as $N$ tends toward $2^n$. A detailed analysis can be found in Appendix A1.

## 6 Limitation and conclusion

In this paper, to address efficient classification under uncertain environments, we propose an adaptive quantum circuit for Dempster's rule of combination (AQC-DRC) to support quantum basic probability and plausibility level decision-making within the framework of quantum evidence theory (QET). The AQC-DRC, as a generalized quantum Bayesian inference method, enables deterministic computation of DRC, ensuring that quantum fusion outcomes in uncertain pattern classification are fully consistent with those of the classical method. Furthermore, it achieves an exponential reduction in computational complexity, positioning it as a promising approach for real-time quantum multisource information fusion. The architecture of the proposed AQC-DRC is conceptually straightforward and highly scalable, which facilitates its practical implementation. It is observed that the quantum basic probability amplitude function in QET can naturally express the quantum amplitude encoding. The quantum basic probability in QET, forming the quantum basic probability distribution, can directly express quantum measurement outcomes for basic probability level decision-making, while the quantum plausibility function in QET can also naturally represent the quantum measurement outcomes for plausibility-level decision-making. These insights not only broaden the understanding of QET, but also provide a more intuitive physical interpretation of quantum amplitude encoding and quantum measurement outcomes.

However, this study revealed that the current version of AQC-DRC is unable to process complex-valued input data, thereby constraining its applicability. Future research will focus on extending AQC-DRC's capabilities through further developments in quantum computing techniques. Notably, Xu et al. [67] proposed a multimodal optimization strategy, which is an excellent strategy to enhance the complementarity of multimodal data and obtain preferable classification results. Consequently, this work holds the potential to incorporate such strategies to establish a trustworthy fusion and decision-making framework for handling heterogeneous data in future applications.

## Acknowledgments

The authors greatly appreciate the reviewers' suggestions and the area chair's encouragement. This research is supported by the National Natural Science Foundation of China (No. 62473067), Xiaomi Young Talents Program, Chongqing Talents: Exceptional Young Talents Project (No. cstc2022ycjh-bgzxm0070), and Chongqing Overseas Scholars Innovation Program (No. cx2022024).

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

# Appendix

## A1 Computational complexity analysis

All algorithms in Table 2 can be divided into two parts: the combination process, which involves obtaining the complete combined evidence or QPl; and the decision-making process, which involves using the obtained evidence or QPl to reach a decision. Assume that there are $n$ elements in the frame of discernment (FOD) and $k$ pieces of evidence, with a total of $N$ focal elements.

1) DRC: When combining two pieces of evidence, the computational complexity of computing any of the focal elements is $O(2^{2n})$. Since there are $N$ focal elements in total, the computational complexity of getting the complete combination result is $O(N2^{2n})$. The total number of combinations is $k - 1$, so the total complexity of combination process is $O(kN2^{2n})$. After the combination process, a decision is made from $n$ elements in the FOD. The computational complexity of the decision-making process is $O(n)$. Thus, the total computational complexity of DRC is $O(kN2^2n)$. ($O(kN2^{2n} + n) = O(kN2^{2n})$)

2) DRC+QPl: The computational complexity required to compute the QPl of a focal element from QBP in the classical process is $O(N)$. In the experimental process we made the decision to use the QPl of elements in FOD, so the computational complexity to compute QPl is $O(nN)$. Since the computational complexity of the DRC is $O(kN2^{2n})$, the total computational complexity of combination process is $O(kN2^{2n})$. ($O(kN2^{2n} + nN) = O(kN2^{2n})$, as $nN < kN2^{2n}$ when $n \geq 2$). The computational complexity of decision-making process is $O(n)$. Thus, the total computational complexity of DRC+QPl is $O(kN2^{2n})$.

3) QC-DRC$_{\text{QBP}}$: The computational complexity of the quantum line part we use the depth of the quantum circuit to denote. The QC-DRC$_{\text{QBP}}$ can only combine two pieces of evidence at a time. For the combination of two evidences the number of Toffoli gates used is $n$, so the computational complexity of the quantum circuit part is $O(n)$. There are $N$ measurements obtained from quantum circuit, each of which requires a normalization process of complexity $O(1)$, so the computational complexity of the normalization part is $O(N)$. The computational complexity of two evidence combinations is $O(n + N)$. In total, $k - 1$ combinations have to be performed, so the computational complexity of combination process is $O(kn + kN)$. The computational complexity of decision-making process is $O(n)$. Thus, the total computational complexity of QC-DRC$_{\text{QBP}}$ is $O(kn + kN)$.

4) QC-DRC$_{\text{QBP}}$+QPl: The computational complexity of QC-DRC$_{\text{QBP}}$ is $O(kn + kN)$, and the computational complexity of the classical process of generating QPl from QBP is $O(nN)$. Therefore, the total computational complexity of combination process is $O(kn + kN + nN)$. The computational complexity of decision-making process is $O(n)$. Thus, the total computational complexity of QC-DRC$_{\text{QBP}}$+QPl is $O(kn + kN + nN)$.

5) AQC-DRC$_{\text{QBP}}$: The AQC-DRC$_{\text{QBP}}$ can combine more than one piece of evidence at a time and the number required Toffoli (CCNOT) gate is $(k-1)n$. Thus, the complexity of the quantum circuit is $O(kn)$. The complexity of the normalization part is consistent with QC-DRC$_{\text{QBP}}$ and is $O(N)$. Thus the total computational complexity of combination process is $O(kn + N)$. The computational complexity of decision-making process is $O(n)$. Thus, the total computational complexity of AQC-DRC$_{\text{QBP}}$ is O(kn+N).

6) AQC-DRC$_{\text{QBP}}$+QPl: The computational complexity of AQC-DRC$_{\text{QBP}}$ is $O(kn + N)$, and the computational complexity of the classical process of generating QPl from QBP is $O(nN)$. Therefore, the total computational complexity of combination process is $O(kn + nN)$. ($O(kn + N + nN) = O(kn + (n + 1)N) = O(kn + nN)$). The computational complexity of decision-making process is $O(n)$. Thus, the total computational complexity of AQC-DRC$_{\text{QBP}}$+QPl is $O(kn + nN)$.

7) AQC-DRC$_{\text{QPl}}$: The algorithm involves measuring $n$ qubits individually, so a total of $n$ quantum circuits are needed to complete it. The computational complexity of each of these quantum circuits is consistent with that of the quantum circuit of AQC-DRC$_{\text{QBP}}$, which is $O(kn)$, and thus its total computational complexity of all quantum circuits is $O(kn^2)$. The computational complexity of decision-making process is $O(n)$. Thus, the total computational complexity of AQC-DRC$_{\text{QPl}}$ is $O(kn^2)$.

## A2   Experimental environment

The classical DRC and DRC+QPl methods were executed using Python 3.8.20 on a Windows 10 system equipped with a 13th Gen Intelő Core i9-13900K CPU (3.00 GHz), 128 GB RAM, and an NVIDIA GeForce RTX 4090 GPU. The quantum methods, including QC-DRCQBP, QC-DRCQBP+QPl, the proposed AQC-DRCQBP, AQC-DRCQBP+QPl, and AQC-DRC$_{QPl}$, were mainly executed on the IonQ quantum simulator (https://ionq.com/).

## A3   Experimental results

Table A1: Comparison of classification accuracies under $\kappa$-th fold of cross-validation for Iris dataset in terms of classical/quantum basic probability level decision-making.

| Method | Shots | $\kappa$-th fold of cross-validation | | | | | Acc |
|---|---|---|---|---|---|---|---|
| | | 1 | 2 | 3 | 4 | 5 | |
| DRC | – | 1.0000 | 0.9000 | 0.9000 | 0.9667 | 0.9667 | 0.9467 |
| QC-DRC$_{QBP}$ | 32 | 0.9000 | 0.9333 | 0.8667 | 0.9000 | 0.9000 | 0.9000 |
| | 64 | 0.9333 | 0.9333 | 0.9333 | 0.9333 | 0.9333 | 0.9333 |
| | 128 | 1.0000 | 0.9333 | 0.9000 | 0.9333 | 0.9667 | 0.9467 |
| | 256 | 1.0000 | 0.9000 | 0.9333 | 0.9000 | 0.9333 | 0.9333 |
| | 512 | 1.0000 | 0.8667 | 0.9333 | 0.9667 | 1.0000 | 0.9533 |
| | 1024 | 1.0000 | 0.9000 | 0.9000 | 0.9333 | 0.9333 | 0.9333 |
| | 2048 | 1.0000 | 0.9333 | 0.8667 | 0.9667 | 0.9667 | 0.9467 |
| | 4096 | 1.0000 | 0.8667 | 0.9000 | 0.9333 | 0.9667 | 0.9333 |
| | 8192 | 1.0000 | 0.9000 | 0.9000 | 0.9667 | 1.0000 | 0.9533 |
| | 16384 | 1.0000 | 0.9000 | 0.9000 | 0.9667 | 0.9667 | 0.9467 |
| AQC-DRC$_{QBP}$ | 32 | 0.9000 | 0.8667 | 0.9000 | 0.9000 | 0.9667 | 0.9067 |
| | 64 | 0.9000 | 0.9333 | 0.8667 | 0.9000 | 0.9667 | 0.9133 |
| | 128 | 0.9667 | 0.9000 | 0.9333 | 0.9333 | 0.9333 | 0.9333 |
| | 256 | 1.0000 | 0.9000 | 0.9000 | 0.9667 | 0.9667 | 0.9467 |
| | 512 | 1.0000 | 0.9000 | 0.9000 | 0.9333 | 0.9667 | 0.9400 |
| | 1024 | 1.0000 | 0.9000 | 0.9333 | 0.9333 | 0.9667 | 0.9467 |
| | 2048 | 1.0000 | 0.8667 | 0.9333 | 0.9667 | 0.9333 | 0.9400 |
| | 4096 | 1.0000 | 0.8667 | 0.9000 | 0.9667 | 0.9333 | 0.9333 |
| | 8192 | 1.0000 | 0.8667 | 0.9000 | 0.9667 | 1.0000 | 0.9467 |
| | 16384 | 1.0000 | 0.8667 | 0.9333 | 0.9667 | 1.0000 | 0.9533 |

Table A2: Comparison of classification accuracies under $\kappa$-th fold of cross-validation for Iris dataset in terms of classical/quantum plausibility level decision-making.

| Method | Shots | $\kappa$-th fold of cross-validation | | | | | Acc |
| --- | --- | --- | --- | --- | --- | --- | --- |
| | | 1 | 2 | 3 | 4 | 5 | |
| DRC+QPl | – | 1.0000 | 0.9000 | 0.9000 | 0.9667 | 0.9667 | 0.9467 |
| QC-DRC$_{QBP}$+QPl | 32 | 0.9000 | 0.7333 | 0.8667 | 0.8000 | 0.9000 | 0.8400 |
| | 64 | 0.8667 | 0.8000 | 0.9000 | 0.8667 | 0.9333 | 0.8733 |
| | 128 | 1.0000 | 0.9333 | 0.9333 | 0.9667 | 0.9333 | 0.9533 |
| | 256 | 1.0000 | 0.9333 | 0.9000 | 0.9667 | 0.9667 | 0.9533 |
| | 512 | 1.0000 | 0.8667 | 0.8667 | 0.9333 | 0.9333 | 0.9200 |
| | 1024 | 1.0000 | 0.9000 | 0.9000 | 0.9333 | 0.9333 | 0.9333 |
| | 2048 | 1.0000 | 0.8667 | 0.9000 | 0.9333 | 1.0000 | 0.9400 |
| | 4096 | 1.0000 | 0.8667 | 0.9000 | 0.9667 | 0.9667 | 0.9400 |
| | 8192 | 1.0000 | 0.9000 | 0.8667 | 0.9667 | 1.0000 | 0.9467 |
| | 16384 | 1.0000 | 0.9000 | 0.9000 | 0.9667 | 0.9667 | 0.9467 |
| AQC-DRC$_{QBP}$+QPl | 32 | 0.9667 | 0.9667 | 0.9000 | 0.9000 | 0.9333 | 0.9333 |
| | 64 | 0.9000 | 0.8333 | 0.9000 | 0.9333 | 0.9667 | 0.9067 |
| | 128 | 1.0000 | 0.9667 | 0.8667 | 0.9000 | 1.0000 | 0.9467 |
| | 256 | 1.0000 | 0.9000 | 0.9333 | 0.9600 | 0.9667 | 0.9500 |
| | 512 | 1.0000 | 0.9000 | 0.9000 | 0.9667 | 0.9667 | 0.9467 |
| | 1024 | 1.0000 | 0.8667 | 0.9000 | 0.9667 | 0.9667 | 0.9400 |
| | 2048 | 1.0000 | 0.9000 | 0.9000 | 0.9667 | 0.9667 | 0.9467 |
| | 4096 | 1.0000 | 0.8667 | 0.8667 | 0.9667 | 1.0000 | 0.9400 |
| | 8192 | 1.0000 | 0.8667 | 0.8667 | 0.9667 | 0.9667 | 0.9333 |
| | 16384 | 1.0000 | 0.9000 | 0.9333 | 0.9667 | 0.9667 | 0.9533 |
| AQC-DRC$_{QPl}$ | 32 | 0.9667 | 0.9333 | 0.8667 | 0.9000 | 0.9333 | 0.9200 |
| | 64 | 1.0000 | 0.9000 | 0.9000 | 0.9667 | 0.9333 | 0.9400 |
| | 128 | 1.0000 | 0.9333 | 0.9000 | 0.9667 | 0.9667 | 0.9533 |
| | 256 | 1.0000 | 0.9333 | 0.9000 | 0.9333 | 0.9667 | 0.9467 |
| | 512 | 1.0000 | 0.9333 | 0.9000 | 0.9333 | 0.9667 | 0.9467 |
| | 1024 | 1.0000 | 0.8667 | 0.9000 | 0.9667 | 0.9667 | 0.9400 |
| | 2048 | 1.0000 | 0.9000 | 0.8667 | 0.9667 | 0.9667 | 0.9400 |
| | 4096 | 1.0000 | 0.9333 | 0.9000 | 0.9667 | 1.0000 | 0.9600 |
| | 8192 | 1.0000 | 0.8667 | 0.9333 | 0.9667 | 0.9667 | 0.9467 |
| | 16384 | 1.0000 | 0.8667 | 0.9000 | 0.9667 | 0.9667 | 0.9400 |

Table A3: Comparison of classification accuracies under $\kappa$-th fold of cross-validation for Abalone dataset in terms of classical/quantum basic probability level decision-making.

| Method | Shots | $\kappa$-th fold of cross-validation | | | | | Acc |
| | | 1 | 2 | 3 | 4 | 5 | |
|---|---|---|---|---|---|---|---|
| DRC | – | 1.0000 | 1.0000 | 0.9767 | 1.0000 | 0.9348 | 0.9823 |
| QC-DRC$_{QBP}$ | 32 | 0.9535 | 1.0000 | 0.9535 | 0.9767 | 0.8261 | 0.9420 |
| | 64 | 0.9535 | 0.9767 | 0.9302 | 1.0000 | 0.8913 | 0.9504 |
| | 128 | 1.0000 | 1.0000 | 0.9535 | 0.9767 | 0.8696 | 0.9600 |
| | 256 | 1.0000 | 0.9767 | 0.9535 | 1.0000 | 0.9130 | 0.9687 |
| | 512 | 1.0000 | 1.0000 | 0.9767 | 1.0000 | 0.9348 | 0.9823 |
| | 1024 | 1.0000 | 1.0000 | 0.9767 | 1.0000 | 0.9565 | 0.9867 |
| | 2048 | 1.0000 | 1.0000 | 0.9767 | 1.0000 | 0.9348 | 0.9823 |
| | 4096 | 1.0000 | 1.0000 | 1.0000 | 1.0000 | 0.9348 | 0.9870 |
| | 8192 | 1.0000 | 1.0000 | 0.9767 | 1.0000 | 0.9348 | 0.9823 |
| | 16384 | 1.0000 | 1.0000 | 0.9767 | 1.0000 | 0.9348 | 0.9823 |
| AQC-DRC$_{QBP}$ | 32 | 1.0000 | 1.0000 | 0.9767 | 1.0000 | 0.9565 | 0.9867 |
| | 64 | 1.0000 | 0.9767 | 0.9767 | 1.0000 | 0.9348 | 0.9777 |
| | 128 | 1.0000 | 1.0000 | 0.9767 | 1.0000 | 0.9348 | 0.9823 |
| | 256 | 1.0000 | 1.0000 | 0.9767 | 1.0000 | 0.9565 | 0.9867 |
| | 512 | 1.0000 | 1.0000 | 0.9767 | 1.0000 | 0.9348 | 0.9823 |
| | 1024 | 1.0000 | 1.0000 | 0.9767 | 1.0000 | 0.9348 | 0.9823 |
| | 2048 | 1.0000 | 1.0000 | 0.9767 | 1.0000 | 0.9348 | 0.9823 |
| | 4096 | 1.0000 | 1.0000 | 0.9767 | 1.0000 | 0.9348 | 0.9823 |
| | 8192 | 1.0000 | 1.0000 | 0.9767 | 1.0000 | 0.9348 | 0.9823 |
| | 16384 | 1.0000 | 1.0000 | 0.9767 | 1.0000 | 0.9348 | 0.9823 |

Table A4: Comparison of classification accuracies under $\kappa$-th fold of cross-validation for Abalone dataset in terms of classical/quantum plausibility level decision-making.

| Method | Shots | $\kappa$-th fold of cross-validation | | | | | Acc |
|---|---|---|---|---|---|---|---|
| | | 1 | 2 | 3 | 4 | 5 | |
| DRC+QPl | – | 1.0000 | 1.0000 | 0.9767 | 1.0000 | 0.9348 | 0.9823 |
| QC-DRC$_{QBP}$+QPl | 32 | 1.0000 | 0.9767 | 1.0000 | 1.0000 | 0.9348 | 0.9823 |
| | 64 | 1.0000 | 1.0000 | 1.0000 | 1.0000 | 0.9348 | 0.9870 |
| | 128 | 1.0000 | 1.0000 | 0.9767 | 1.0000 | 0.9348 | 0.9823 |
| | 256 | 1.0000 | 1.0000 | 0.9767 | 1.0000 | 0.9565 | 0.9867 |
| | 512 | 1.0000 | 1.0000 | 0.9767 | 1.0000 | 0.9565 | 0.9867 |
| | 1024 | 1.0000 | 1.0000 | 0.9767 | 1.0000 | 0.9348 | 0.9823 |
| | 2048 | 1.0000 | 1.0000 | 0.9767 | 1.0000 | 0.9348 | 0.9823 |
| | 4096 | 1.0000 | 1.0000 | 0.9767 | 1.0000 | 0.9348 | 0.9823 |
| | 8192 | 1.0000 | 1.0000 | 0.9767 | 1.0000 | 0.9348 | 0.9823 |
| | 16384 | 1.0000 | 1.0000 | 0.9767 | 1.0000 | 0.9348 | 0.9823 |
| AQC-DRC$_{QBP}$+QPl | 32 | 0.9070 | 1.0000 | 0.9535 | 0.9070 | 0.7609 | 0.9057 |
| | 64 | 0.9535 | 0.9767 | 0.9535 | 1.0000 | 0.8478 | 0.9463 |
| | 128 | 1.0000 | 0.9767 | 0.9535 | 0.9767 | 0.8913 | 0.9597 |
| | 256 | 1.0000 | 1.0000 | 0.9767 | 1.0000 | 0.9348 | 0.9823 |
| | 512 | 1.0000 | 0.9767 | 0.9767 | 1.0000 | 0.9348 | 0.9777 |
| | 1024 | 1.0000 | 1.0000 | 1.0000 | 1.0000 | 0.9565 | 0.9913 |
| | 2048 | 1.0000 | 1.0000 | 0.9767 | 1.0000 | 0.9348 | 0.9823 |
| | 4096 | 1.0000 | 1.0000 | 0.9767 | 1.0000 | 0.9348 | 0.9823 |
| | 8192 | 1.0000 | 1.0000 | 0.9767 | 1.0000 | 0.9565 | 0.9867 |
| | 16384 | 1.0000 | 1.0000 | 0.9767 | 1.0000 | 0.9348 | 0.9823 |
| AQC-DRC$_{QPl}$ | 32 | 0.9767 | 0.9767 | 0.9302 | 0.9070 | 0.8261 | 0.9234 |
| | 64 | 1.0000 | 0.9767 | 0.9767 | 1.0000 | 0.8261 | 0.9559 |
| | 128 | 1.0000 | 0.9767 | 0.9767 | 1.0000 | 0.9565 | 0.9820 |
| | 256 | 1.0000 | 0.9767 | 0.9767 | 1.0000 | 0.9348 | 0.9777 |
| | 512 | 1.0000 | 0.9767 | 0.9767 | 1.0000 | 0.9348 | 0.9777 |
| | 1024 | 1.0000 | 0.9767 | 1.0000 | 1.0000 | 0.9348 | 0.9823 |
| | 2048 | 1.0000 | 0.9767 | 0.9767 | 1.0000 | 0.9348 | 0.9777 |
| | 4096 | 1.0000 | 1.0000 | 0.9767 | 1.0000 | 0.9348 | 0.9823 |
| | 8192 | 1.0000 | 1.0000 | 0.9767 | 1.0000 | 0.9348 | 0.9823 |
| | 16384 | 1.0000 | 1.0000 | 0.9767 | 1.0000 | 0.9348 | 0.9823 |

Table A5: Comparison of classification accuracies under $\kappa$-th fold of cross-validation for Knowledge dataset in terms of classical/quantum basic probability level decision-making.

| Method | Shots | $\kappa$-th fold of cross-validation | | | | | Acc |
| | | 1 | 2 | 3 | 4 | 5 | |
|---|---|---|---|---|---|---|---|
| DRC | – | 0.8667 | 0.8312 | 0.8267 | 0.8592 | 0.8861 | 0.8539 |
| QC-DRC$_{QBP}$ | 32 | 0.6267 | 0.5455 | 0.6133 | 0.5775 | 0.6582 | 0.6042 |
| | 64 | 0.8000 | 0.6883 | 0.6667 | 0.7324 | 0.6456 | 0.7066 |
| | 128 | 0.7467 | 0.8052 | 0.6800 | 0.7887 | 0.7595 | 0.7560 |
| | 256 | 0.8533 | 0.7792 | 0.7733 | 0.8028 | 0.8481 | 0.8114 |
| | 512 | 0.8667 | 0.7922 | 0.8400 | 0.8169 | 0.8354 | 0.8302 |
| | 1024 | 0.8667 | 0.7922 | 0.8667 | 0.8451 | 0.8734 | 0.8488 |
| | 2048 | 0.8667 | 0.8052 | 0.8400 | 0.8451 | 0.8861 | 0.8486 |
| | 4096 | 0.8667 | 0.8312 | 0.8000 | 0.8592 | 0.8987 | 0.8511 |
| | 8192 | 0.8800 | 0.8182 | 0.8400 | 0.8873 | 0.8861 | 0.8623 |
| | 16384 | 0.8667 | 0.8442 | 0.8267 | 0.8310 | 0.8861 | 0.8509 |
| AQC-DRC$_{QBP}$ | 32 | 0.6400 | 0.5844 | 0.6800 | 0.5634 | 0.6329 | 0.6201 |
| | 64 | 0.6800 | 0.6883 | 0.6000 | 0.7183 | 0.6962 | 0.6766 |
| | 128 | 0.6933 | 0.7532 | 0.7333 | 0.7183 | 0.7468 | 0.7290 |
| | 256 | 0.7467 | 0.7532 | 0.7600 | 0.7606 | 0.8861 | 0.7813 |
| | 512 | 0.8667 | 0.8052 | 0.8133 | 0.8169 | 0.8101 | 0.8224 |
| | 1024 | 0.8667 | 0.7532 | 0.7733 | 0.8592 | 0.8987 | 0.8302 |
| | 2048 | 0.8800 | 0.8182 | 0.8400 | 0.8169 | 0.8734 | 0.8457 |
| | 4096 | 0.8800 | 0.8182 | 0.8267 | 0.8592 | 0.8734 | 0.8515 |
| | 8192 | 0.8667 | 0.8052 | 0.8267 | 0.8592 | 0.8734 | 0.8462 |
| | 16384 | 0.8800 | 0.8312 | 0.8267 | 0.8451 | 0.8861 | 0.8538 |

Table A6: Comparison of classification accuracies under $\kappa$-th fold of cross-validation for Knowledge dataset in terms of classical/quantum plausibility level decision-making.

| Method | Shots | $\kappa$-th fold of cross-validation | | | | | Acc |
| | | 1 | 2 | 3 | 4 | 5 | |
|---|---|---|---|---|---|---|---|
| DRC+QPl | – | 0.8667 | 0.8571 | 0.8267 | 0.8592 | 0.8861 | 0.8591 |
| QC-DRC$_{QBP}$+QPl | 32 | 0.6267 | 0.5844 | 0.6267 | 0.6479 | 0.7089 | 0.6389 |
| | 64 | 0.7333 | 0.7013 | 0.6133 | 0.6901 | 0.7342 | 0.6945 |
| | 128 | 0.7600 | 0.7403 | 0.7067 | 0.7887 | 0.7215 | 0.7434 |
| | 256 | 0.7467 | 0.7403 | 0.7867 | 0.8028 | 0.8481 | 0.7849 |
| | 512 | 0.8267 | 0.8442 | 0.8133 | 0.8592 | 0.8354 | 0.8358 |
| | 1024 | 0.8533 | 0.8442 | 0.8400 | 0.8451 | 0.8481 | 0.8461 |
| | 2048 | 0.8667 | 0.8182 | 0.8400 | 0.8451 | 0.8734 | 0.8487 |
| | 4096 | 0.8667 | 0.8442 | 0.8533 | 0.8732 | 0.8861 | 0.8647 |
| | 8192 | 0.8533 | 0.8312 | 0.8533 | 0.8732 | 0.8861 | 0.8594 |
| | 16384 | 0.8667 | 0.8571 | 0.8267 | 0.8592 | 0.8861 | 0.8591 |
| AQC-DRC$_{QBP}$+QPl | 32 | 0.6000 | 0.6494 | 0.5600 | 0.5915 | 0.6203 | 0.6042 |
| | 64 | 0.7200 | 0.7403 | 0.6533 | 0.7183 | 0.7468 | 0.7157 |
| | 128 | 0.7467 | 0.7792 | 0.7333 | 0.8310 | 0.8354 | 0.7851 |
| | 256 | 0.8133 | 0.7532 | 0.7733 | 0.7746 | 0.8481 | 0.7925 |
| | 512 | 0.8267 | 0.8182 | 0.8400 | 0.8169 | 0.8734 | 0.8350 |
| | 1024 | 0.8667 | 0.8442 | 0.8400 | 0.8873 | 0.8861 | 0.8648 |
| | 2048 | 0.8400 | 0.8571 | 0.8400 | 0.8592 | 0.8861 | 0.8565 |
| | 4096 | 0.8533 | 0.8442 | 0.8267 | 0.8451 | 0.8861 | 0.8511 |
| | 8192 | 0.8667 | 0.8442 | 0.8400 | 0.8732 | 0.8861 | 0.8620 |
| | 16384 | 0.8667 | 0.8571 | 0.8267 | 0.8732 | 0.8861 | 0.8620 |
| AQC-DRC$_{QPl}$ | 32 | 0.6133 | 0.6753 | 0.6533 | 0.7042 | 0.6203 | 0.6533 |
| | 64 | 0.7067 | 0.7403 | 0.6933 | 0.7324 | 0.7595 | 0.7264 |
| | 128 | 0.8133 | 0.7403 | 0.8000 | 0.7324 | 0.7722 | 0.7716 |
| | 256 | 0.8133 | 0.7662 | 0.8133 | 0.8169 | 0.8734 | 0.8166 |
| | 512 | 0.8667 | 0.7662 | 0.8000 | 0.8310 | 0.8608 | 0.8249 |
| | 1024 | 0.8533 | 0.8182 | 0.8267 | 0.8732 | 0.8861 | 0.8515 |
| | 2048 | 0.8533 | 0.8312 | 0.8133 | 0.8732 | 0.8987 | 0.8540 |
| | 4096 | 0.8533 | 0.8442 | 0.8267 | 0.8592 | 0.8734 | 0.8513 |
| | 8192 | 0.8667 | 0.8312 | 0.8133 | 0.8592 | 0.8861 | 0.8513 |
| | 16384 | 0.8667 | 0.8571 | 0.8133 | 0.8592 | 0.8861 | 0.8565 |

Table A7: Comparison of Dif and StD metrics under $\kappa$-fold cross-validation for Iris dataset in terms of quantum basic probability level decision-making.

| Method | Shots | $\kappa$-th fold of cross-validation | | | | | Dif | StD |
|--------|-------|--------|--------|--------|--------|--------|--------|--------|
| | | 1 | 2 | 3 | 4 | 5 | | |
| QC-DRC$_{QBP}$ | 32 | 0.1000 | 0.1000 | 0.1000 | 0.0667 | 0.1333 | 0.1000 | 0.0211 |
| | 64 | 0.0667 | 0.1000 | 0.0333 | 0.0333 | 0.1000 | 0.0667 | 0.0298 |
| | 128 | 0.0000 | 0.1000 | 0.0667 | 0.0333 | 0.0667 | 0.0533 | 0.0340 |
| | 256 | 0.0000 | 0.0000 | 0.0333 | 0.0667 | 0.0333 | 0.0267 | 0.0249 |
| | 512 | 0.0000 | 0.0333 | 0.0333 | 0.0000 | 0.0333 | 0.0200 | 0.0163 |
| | 1024 | 0.0000 | 0.0667 | 0.0000 | 0.0333 | 0.0333 | 0.0267 | 0.0249 |
| | 2048 | 0.0000 | 0.0333 | 0.0333 | 0.0000 | 0.0000 | 0.0133 | 0.0163 |
| | 4096 | 0.0000 | 0.0333 | 0.0000 | 0.0333 | 0.0000 | 0.0133 | 0.0163 |
| | 8192 | 0.0000 | 0.0000 | 0.0000 | 0.0000 | 0.0333 | 0.0067 | 0.0133 |
| | 16384 | 0.0000 | 0.0000 | 0.0000 | 0.0000 | 0.0000 | 0.0000 | 0.0000 |
| AQC-DRC$_{QBP}$ | 32 | 0.1000 | 0.1667 | 0.0000 | 0.0667 | 0.0667 | 0.0800 | 0.0542 |
| | 64 | 0.1000 | 0.1000 | 0.0333 | 0.0667 | 0.0667 | 0.0733 | 0.0249 |
| | 128 | 0.0333 | 0.0667 | 0.0333 | 0.0333 | 0.1000 | 0.0533 | 0.0267 |
| | 256 | 0.0000 | 0.0000 | 0.0667 | 0.0000 | 0.0000 | 0.0133 | 0.0267 |
| | 512 | 0.0000 | 0.0000 | 0.0000 | 0.0333 | 0.0000 | 0.0067 | 0.0133 |
| | 1024 | 0.0000 | 0.0667 | 0.0333 | 0.0333 | 0.0000 | 0.0267 | 0.0249 |
| | 2048 | 0.0000 | 0.0333 | 0.0333 | 0.0000 | 0.0333 | 0.0200 | 0.0163 |
| | 4096 | 0.0000 | 0.0333 | 0.0667 | 0.0000 | 0.0333 | 0.0267 | 0.0249 |
| | 8192 | 0.0000 | 0.0333 | 0.0000 | 0.0000 | 0.0333 | 0.0133 | 0.0163 |
| | 16384 | 0.0000 | 0.0333 | 0.0333 | 0.0000 | 0.0333 | 0.0200 | 0.0163 |

Table A8: Comparison of Dif and StD metrics under $\kappa$-fold cross-validation for Iris dataset in terms of quantum plausibility level decision-making.

| Method | Shots | $\kappa$-th fold of cross-validation | | | | | Dif | StD |
| | | 1 | 2 | 3 | 4 | 5 | | |
|---|---|---|---|---|---|---|---|---|
| QC-DRC$_{QBP}$+QPl | 32 | 0.1000 | 0.1667 | 0.1000 | 0.1667 | 0.1333 | 0.1333 | 0.0298 |
| | 64 | 0.1333 | 0.1667 | 0.0667 | 0.1000 | 0.1000 | 0.1133 | 0.0340 |
| | 128 | 0.0000 | 0.0333 | 0.0333 | 0.0000 | 0.1000 | 0.0333 | 0.0365 |
| | 256 | 0.0000 | 0.0333 | 0.0667 | 0.0000 | 0.0667 | 0.0333 | 0.0298 |
| | 512 | 0.0000 | 0.0333 | 0.0333 | 0.0333 | 0.0333 | 0.0267 | 0.0133 |
| | 1024 | 0.0000 | 0.0000 | 0.0667 | 0.0333 | 0.0333 | 0.0267 | 0.0249 |
| | 2048 | 0.0000 | 0.0333 | 0.0000 | 0.0333 | 0.0333 | 0.0200 | 0.0163 |
| | 4096 | 0.0000 | 0.0333 | 0.0000 | 0.0000 | 0.0000 | 0.0067 | 0.0133 |
| | 8192 | 0.0000 | 0.0000 | 0.0333 | 0.0000 | 0.0333 | 0.0133 | 0.0163 |
| | 16384 | 0.0000 | 0.0000 | 0.0000 | 0.0000 | 0.0000 | 0.0000 | 0.0000 |
| AQC-DRC$_{QBP}$+QPl | 32 | 0.0333 | 0.0667 | 0.0667 | 0.0667 | 0.1000 | 0.0667 | 0.0211 |
| | 64 | 0.1000 | 0.0667 | 0.0667 | 0.0333 | 0.0000 | 0.0533 | 0.0340 |
| | 128 | 0.0000 | 0.0667 | 0.0333 | 0.0667 | 0.0333 | 0.0400 | 0.0249 |
| | 256 | 0.0000 | 0.0667 | 0.0333 | 0.0000 | 0.0000 | 0.0200 | 0.0267 |
| | 512 | 0.0000 | 0.0667 | 0.0667 | 0.0000 | 0.0000 | 0.0267 | 0.0327 |
| | 1024 | 0.0000 | 0.0333 | 0.0667 | 0.0000 | 0.0000 | 0.0200 | 0.0267 |
| | 2048 | 0.0000 | 0.0000 | 0.0000 | 0.0000 | 0.0000 | 0.0000 | 0.0000 |
| | 4096 | 0.0000 | 0.0333 | 0.0333 | 0.0000 | 0.0333 | 0.0200 | 0.0163 |
| | 8192 | 0.0000 | 0.0333 | 0.0333 | 0.0000 | 0.0000 | 0.0133 | 0.0163 |
| | 16384 | 0.0000 | 0.0000 | 0.0333 | 0.0000 | 0.0000 | 0.0067 | 0.0133 |
| AQC-DRC$_{QPl}$ | 32 | 0.0333 | 0.1000 | 0.0333 | 0.0667 | 0.0333 | 0.0533 | 0.0267 |
| | 64 | 0.0000 | 0.1333 | 0.0667 | 0.0000 | 0.1000 | 0.0600 | 0.0533 |
| | 128 | 0.0000 | 0.1000 | 0.0667 | 0.0000 | 0.0000 | 0.0333 | 0.0422 |
| | 256 | 0.0000 | 0.0333 | 0.0000 | 0.0333 | 0.0000 | 0.0133 | 0.0163 |
| | 512 | 0.0000 | 0.0333 | 0.0000 | 0.0333 | 0.0000 | 0.0133 | 0.0163 |
| | 1024 | 0.0000 | 0.0333 | 0.0000 | 0.0000 | 0.0000 | 0.0067 | 0.0133 |
| | 2048 | 0.0000 | 0.0000 | 0.0333 | 0.0000 | 0.0000 | 0.0067 | 0.0133 |
| | 4096 | 0.0000 | 0.0333 | 0.0000 | 0.0000 | 0.0333 | 0.0133 | 0.0163 |
| | 8192 | 0.0000 | 0.0333 | 0.0333 | 0.0000 | 0.0000 | 0.0133 | 0.0163 |
| | 16384 | 0.0000 | 0.0333 | 0.0000 | 0.0000 | 0.0000 | 0.0067 | 0.0133 |

Table A9: Comparison of Dif and StD metrics under $\kappa$-fold cross-validation for Abalone dataset in terms of quantum basic probability level decision-making.

| Method | Shots | $\kappa$-th fold of cross-validation | | | | | Dif | StD |
|--------|-------|--------|--------|--------|--------|--------|--------|--------|
| | | 1 | 2 | 3 | 4 | 5 | | |
| QC-DRC$_{QBP}$ | 32 | 0.0465 | 0.0000 | 0.0233 | 0.0233 | 0.1522 | 0.0490 | 0.0536 |
| | 64 | 0.0465 | 0.0233 | 0.0465 | 0.0000 | 0.0870 | 0.0406 | 0.0289 |
| | 128 | 0.0000 | 0.0000 | 0.0233 | 0.0233 | 0.0652 | 0.0223 | 0.0238 |
| | 256 | 0.0000 | 0.0233 | 0.0233 | 0.0000 | 0.0217 | 0.0137 | 0.0112 |
| | 512 | 0.0000 | 0.0000 | 0.0000 | 0.0000 | 0.0000 | 0.0000 | 0.0000 |
| | 1024 | 0.0000 | 0.0000 | 0.0000 | 0.0000 | 0.0217 | 0.0043 | 0.0087 |
| | 2048 | 0.0000 | 0.0000 | 0.0000 | 0.0000 | 0.0000 | 0.0000 | 0.0000 |
| | 4096 | 0.0000 | 0.0000 | 0.0233 | 0.0000 | 0.0000 | 0.0047 | 0.0093 |
| | 8192 | 0.0000 | 0.0000 | 0.0000 | 0.0000 | 0.0000 | 0.0000 | 0.0000 |
| | 16384 | 0.0000 | 0.0000 | 0.0000 | 0.0000 | 0.0000 | 0.0000 | 0.0000 |
| AQC-DRC$_{QBP}$ | 32 | 0.0000 | 0.0000 | 0.0000 | 0.0000 | 0.0217 | 0.0043 | 0.0087 |
| | 64 | 0.0000 | 0.0233 | 0.0000 | 0.0000 | 0.0000 | 0.0047 | 0.0093 |
| | 128 | 0.0000 | 0.0000 | 0.0000 | 0.0000 | 0.0000 | 0.0000 | 0.0000 |
| | 256 | 0.0000 | 0.0000 | 0.0000 | 0.0000 | 0.0217 | 0.0043 | 0.0087 |
| | 512 | 0.0000 | 0.0000 | 0.0000 | 0.0000 | 0.0000 | 0.0000 | 0.0000 |
| | 1024 | 0.0000 | 0.0000 | 0.0000 | 0.0000 | 0.0000 | 0.0000 | 0.0000 |
| | 2048 | 0.0000 | 0.0000 | 0.0000 | 0.0000 | 0.0000 | 0.0000 | 0.0000 |
| | 4096 | 0.0000 | 0.0000 | 0.0000 | 0.0000 | 0.0000 | 0.0000 | 0.0000 |
| | 8192 | 0.0000 | 0.0000 | 0.0000 | 0.0000 | 0.0000 | 0.0000 | 0.0000 |
| | 16384 | 0.0000 | 0.0000 | 0.0000 | 0.0000 | 0.0000 | 0.0000 | 0.0000 |

Table A10: Comparison of Dif and StD metrics under $\kappa$-fold cross-validation for Abalone dataset in terms of quantum plausibility level decision-making.

| Method | Shots | $\kappa$-th fold of cross-validation | | | | | Dif | StD |
|---|---|---|---|---|---|---|---|---|
| | | 1 | 2 | 3 | 4 | 5 | | |
| QC-DRC$_{QBP}$+QPl | 32 | 0.0000 | 0.0233 | 0.0233 | 0.0000 | 0.0000 | 0.0093 | 0.0114 |
| | 64 | 0.0000 | 0.0000 | 0.0233 | 0.0000 | 0.0000 | 0.0047 | 0.0093 |
| | 128 | 0.0000 | 0.0000 | 0.0000 | 0.0000 | 0.0000 | 0.0000 | 0.0000 |
| | 256 | 0.0000 | 0.0000 | 0.0000 | 0.0000 | 0.0217 | 0.0043 | 0.0087 |
| | 512 | 0.0000 | 0.0000 | 0.0000 | 0.0000 | 0.0217 | 0.0043 | 0.0087 |
| | 1024 | 0.0000 | 0.0000 | 0.0000 | 0.0000 | 0.0000 | 0.0000 | 0.0000 |
| | 2048 | 0.0000 | 0.0000 | 0.0000 | 0.0000 | 0.0000 | 0.0000 | 0.0000 |
| | 4096 | 0.0000 | 0.0000 | 0.0000 | 0.0000 | 0.0000 | 0.0000 | 0.0000 |
| | 8192 | 0.0000 | 0.0000 | 0.0000 | 0.0000 | 0.0000 | 0.0000 | 0.0000 |
| | 16384 | 0.0000 | 0.0000 | 0.0000 | 0.0000 | 0.0000 | 0.0000 | 0.0000 |
| AQC-DRC$_{QBP}$+QPl | 32 | 0.0930 | 0.0000 | 0.0233 | 0.0930 | 0.1739 | 0.0766 | 0.0612 |
| | 64 | 0.0465 | 0.0233 | 0.0233 | 0.0000 | 0.1304 | 0.0447 | 0.0453 |
| | 128 | 0.0000 | 0.0233 | 0.0233 | 0.0233 | 0.0435 | 0.0226 | 0.0138 |
| | 256 | 0.0000 | 0.0000 | 0.0000 | 0.0000 | 0.0000 | 0.0000 | 0.0000 |
| | 512 | 0.0000 | 0.0233 | 0.0000 | 0.0000 | 0.0000 | 0.0047 | 0.0093 |
| | 1024 | 0.0000 | 0.0000 | 0.0233 | 0.0000 | 0.0217 | 0.0090 | 0.0110 |
| | 2048 | 0.0000 | 0.0000 | 0.0000 | 0.0000 | 0.0000 | 0.0000 | 0.0000 |
| | 4096 | 0.0000 | 0.0000 | 0.0000 | 0.0000 | 0.0000 | 0.0000 | 0.0000 |
| | 8192 | 0.0000 | 0.0000 | 0.0000 | 0.0000 | 0.0217 | 0.0043 | 0.0087 |
| | 16384 | 0.0000 | 0.0000 | 0.0000 | 0.0000 | 0.0000 | 0.0000 | 0.0000 |
| AQC-DRC$_{QPl}$ | 32 | 0.0233 | 0.0233 | 0.0465 | 0.0930 | 0.1522 | 0.0676 | 0.0493 |
| | 64 | 0.0000 | 0.0233 | 0.0465 | 0.0000 | 0.1087 | 0.0357 | 0.0404 |
| | 128 | 0.0000 | 0.0233 | 0.0000 | 0.0000 | 0.0217 | 0.0090 | 0.0110 |
| | 256 | 0.0000 | 0.0233 | 0.0465 | 0.0000 | 0.0000 | 0.0140 | 0.0186 |
| | 512 | 0.0000 | 0.0233 | 0.0000 | 0.0000 | 0.0000 | 0.0047 | 0.0093 |
| | 1024 | 0.0000 | 0.0233 | 0.0233 | 0.0000 | 0.0000 | 0.0093 | 0.0114 |
| | 2048 | 0.0000 | 0.0233 | 0.0000 | 0.0000 | 0.0000 | 0.0047 | 0.0093 |
| | 4096 | 0.0000 | 0.0000 | 0.0000 | 0.0000 | 0.0000 | 0.0000 | 0.0000 |
| | 8192 | 0.0000 | 0.0000 | 0.0000 | 0.0000 | 0.0000 | 0.0000 | 0.0000 |
| | 16384 | 0.0000 | 0.0000 | 0.0000 | 0.0000 | 0.0000 | 0.0000 | 0.0000 |

Table A11: Comparison of Dif and StD metrics under $\kappa$-fold cross-validation for Knowledge dataset in terms of quantum basic probability level decision-making.

| Method | Shots | $\kappa$-th fold of cross-validation | | | | | Dif | StD |
|---|---|---|---|---|---|---|---|---|
| | | 1 | 2 | 3 | 4 | 5 | | |
| QC-DRC$_{QBP}$ | 32 | 0.3200 | 0.3117 | 0.2933 | 0.3380 | 0.2785 | 0.3083 | 0.0207 |
| | 64 | 0.1200 | 0.2727 | 0.2133 | 0.1831 | 0.2911 | 0.2161 | 0.0619 |
| | 128 | 0.1467 | 0.1039 | 0.2267 | 0.1549 | 0.1772 | 0.1619 | 0.0402 |
| | 256 | 0.0400 | 0.0779 | 0.1067 | 0.1408 | 0.0633 | 0.0857 | 0.0350 |
| | 512 | 0.0000 | 0.0649 | 0.0400 | 0.0704 | 0.0506 | 0.0452 | 0.0250 |
| | 1024 | 0.0000 | 0.0649 | 0.0667 | 0.0423 | 0.0380 | 0.0424 | 0.0241 |
| | 2048 | 0.0000 | 0.0779 | 0.0667 | 0.0141 | 0.0000 | 0.0317 | 0.0337 |
| | 4096 | 0.0000 | 0.0519 | 0.0267 | 0.0563 | 0.0127 | 0.0295 | 0.0218 |
| | 8192 | 0.0133 | 0.0130 | 0.0400 | 0.0563 | 0.0000 | 0.0245 | 0.0205 |
| | 16384 | 0.0000 | 0.0130 | 0.0000 | 0.0563 | 0.0000 | 0.0139 | 0.0218 |
| AQC-DRC$_{QBP}$ | 32 | 0.3333 | 0.3766 | 0.2800 | 0.3803 | 0.3038 | 0.3348 | 0.0395 |
| | 64 | 0.2667 | 0.2208 | 0.3067 | 0.1690 | 0.2658 | 0.2458 | 0.0470 |
| | 128 | 0.1733 | 0.1818 | 0.1467 | 0.1690 | 0.1646 | 0.1671 | 0.0117 |
| | 256 | 0.1200 | 0.2078 | 0.1200 | 0.1268 | 0.0253 | 0.1200 | 0.0578 |
| | 512 | 0.0267 | 0.0779 | 0.0400 | 0.1268 | 0.0759 | 0.0695 | 0.0349 |
| | 1024 | 0.0267 | 0.0779 | 0.1067 | 0.0563 | 0.0380 | 0.0611 | 0.0286 |
| | 2048 | 0.0133 | 0.0130 | 0.0133 | 0.0423 | 0.0127 | 0.0189 | 0.0117 |
| | 4096 | 0.0133 | 0.0390 | 0.0267 | 0.0282 | 0.0127 | 0.0240 | 0.0099 |
| | 8192 | 0.0000 | 0.0779 | 0.0267 | 0.0000 | 0.0127 | 0.0234 | 0.0290 |
| | 16384 | 0.0133 | 0.0260 | 0.0000 | 0.0141 | 0.0000 | 0.0107 | 0.0098 |

Table A12: Comparison of Dif and StD metrics under $\kappa$-fold cross-validation for Knowledge dataset in terms of quantum plausibility level decision-making.

| Method | Shots | $\kappa$-th fold of cross-validation | | | | | Dif | StD |
| | | 1 | 2 | 3 | 4 | 5 | | |
|---|---|---|---|---|---|---|---|---|
| QC-DRC$_{QBP}$+QPl | 32 | 0.2933 | 0.3247 | 0.3600 | 0.2958 | 0.3038 | 0.3155 | 0.0248 |
| | 64 | 0.2133 | 0.2078 | 0.2933 | 0.2254 | 0.2025 | 0.2285 | 0.0333 |
| | 128 | 0.1333 | 0.1429 | 0.1733 | 0.1268 | 0.1899 | 0.1532 | 0.0243 |
| | 256 | 0.1200 | 0.1688 | 0.1200 | 0.1408 | 0.0886 | 0.1277 | 0.0265 |
| | 512 | 0.0400 | 0.0649 | 0.0933 | 0.0845 | 0.0759 | 0.0717 | 0.0184 |
| | 1024 | 0.0133 | 0.0390 | 0.0933 | 0.0423 | 0.0380 | 0.0452 | 0.0262 |
| | 2048 | 0.0267 | 0.0390 | 0.0400 | 0.0423 | 0.0127 | 0.0321 | 0.0111 |
| | 4096 | 0.0000 | 0.0130 | 0.0267 | 0.0141 | 0.0000 | 0.0107 | 0.0100 |
| | 8192 | 0.0133 | 0.0260 | 0.0533 | 0.0141 | 0.0000 | 0.0213 | 0.0180 |
| | 16384 | 0.0000 | 0.0000 | 0.0533 | 0.0282 | 0.0000 | 0.0163 | 0.0215 |
| AQC-DRC$_{QBP}$+QPl | 32 | 0.2667 | 0.2857 | 0.3467 | 0.2958 | 0.3165 | 0.3023 | 0.0274 |
| | 64 | 0.1733 | 0.1688 | 0.2800 | 0.1972 | 0.1899 | 0.2018 | 0.0404 |
| | 128 | 0.1200 | 0.1299 | 0.1467 | 0.0563 | 0.1266 | 0.1159 | 0.0310 |
| | 256 | 0.0533 | 0.1558 | 0.1067 | 0.0845 | 0.0886 | 0.0978 | 0.0337 |
| | 512 | 0.0400 | 0.1169 | 0.0400 | 0.0986 | 0.0127 | 0.0616 | 0.0394 |
| | 1024 | 0.0000 | 0.0130 | 0.0400 | 0.0282 | 0.0253 | 0.0213 | 0.0137 |
| | 2048 | 0.0267 | 0.0000 | 0.0400 | 0.0563 | 0.0253 | 0.0297 | 0.0186 |
| | 4096 | 0.0133 | 0.0130 | 0.0800 | 0.0423 | 0.0000 | 0.0297 | 0.0287 |
| | 8192 | 0.0000 | 0.0390 | 0.0400 | 0.0141 | 0.0000 | 0.0186 | 0.0178 |
| | 16384 | 0.0000 | 0.0000 | 0.0533 | 0.0141 | 0.0000 | 0.0135 | 0.0207 |
| AQC-DRC$_{QPl}$ | 32 | 0.3067 | 0.3117 | 0.3867 | 0.2113 | 0.3418 | 0.3116 | 0.0577 |
| | 64 | 0.1600 | 0.1948 | 0.1867 | 0.2113 | 0.2278 | 0.1961 | 0.0229 |
| | 128 | 0.0800 | 0.1429 | 0.0800 | 0.1831 | 0.1139 | 0.1200 | 0.0393 |
| | 256 | 0.0533 | 0.0909 | 0.0400 | 0.0704 | 0.0380 | 0.0585 | 0.0199 |
| | 512 | 0.0533 | 0.1169 | 0.0533 | 0.0563 | 0.0506 | 0.0661 | 0.0255 |
| | 1024 | 0.0400 | 0.0390 | 0.0800 | 0.0141 | 0.0000 | 0.0346 | 0.0273 |
| | 2048 | 0.0133 | 0.0260 | 0.0667 | 0.0141 | 0.0127 | 0.0265 | 0.0207 |
| | 4096 | 0.0133 | 0.0130 | 0.0000 | 0.0282 | 0.0380 | 0.0185 | 0.0132 |
| | 8192 | 0.0000 | 0.0260 | 0.0133 | 0.0000 | 0.0000 | 0.0079 | 0.0104 |
| | 16384 | 0.0000 | 0.0000 | 0.0400 | 0.0282 | 0.0000 | 0.0136 | 0.0171 |

