# OpenReview forum: "An Adaptive Quantum Circuit of Dempster's Rule of Combination for Uncertain Pattern Classification"
_NeurIPS.cc/2025/Conference — NeurIPS 2025 poster_

### Official Review · Reviewer_rA57 · 2025-06-21

**Clarity:** 4
**Significance:** 3
**Originality:** 4
**Rating:** 5
**Confidence:** 5

**Summary:**

This paper proposes an Adaptive Quantum Circuit for Dempster's Rule of Combination (AQC-DRC) to address uncertainty in pattern classification. The method leverages quantum evidence theory (QET) to provide efficient evidence fusion while maintaining consistency with classical Dempster-Shafer theory results. The AQC-DRC achieves exponential reductions in computational complexity and supports decision-making based on both quantum basic probability and plausibility levels. The authors present rigorous theoretical derivations, experimental evaluations on datasets such as Iris and Abalone, and a detailed comparison with classical and quantum methods.

**Questions:**

1. Why the quantum basic probability amplitude function in QET can be naturally utilized to express the quantum amplitude encoding?
2. What’s different between the quantum basic probability level decision-making and quantum plausibility level decision-making?
3. Figure A1 is a main flowchart of the adaptive quantum circuit for Dempster’s rule of combination. It should be placed in the main text.

**Ethical Concerns:**

["NO or VERY MINOR ethics concerns only"]

**Final Justification:**

The paper presents AQC-DRC within QET, delivering deterministic fusion exactly equivalent to classical DRC while achieving exponential reductions in evidence-combination complexity and supporting both QBP- and QPl-level decisions. The rebuttal convincingly clarifies that QBPA naturally fits amplitude encoding (normalization-consistent) and rigorously maps measurement outcomes to QBP and QPl, with clear guidance on use cases and complexity trade-offs. Experiments on Iris and Abalone substantiate correctness and efficiency; the authors will streamline the Preliminaries and move the main flowchart into the text to improve clarity. Therefore, I would like to keep my positive score.

**Limitations:**

yes

**Paper Formatting Concerns:**

No paper formatting concerns.

**Quality:**

3

**Strengths And Weaknesses:**

Strengths
In this paper, the authors proposed an Adaptive Quantum Circuit for Dempster’s Rule of Combination (AQC-DRC) to address efficient classification under uncertain environments by leveraging the advanced computational power of quantum computing. The AQC-DRC is developed within the framework of quantum evidence theory (QET) and facilitates decision-making based on quantum basic probability and plausibility levels, which is a generalized Bayesian inference method. The main strengths of this paper are summarized as follows:
1. The AQC-DRC provides a deterministic computation of DRC, ensuring that quantum fusion outcomes in uncertain pattern classification are exactly aligned with those of the classical method, while simultaneously achieving exponential reductions in the computational complexity of evidence combination and significantly improving fusion efficiency.
2. It is founded that the quantum basic probability amplitude function in QET, as a generalized quantum probability amplitude, can be naturally utilized to express the quantum amplitude encoding. In addition, the quantum basic probability in QET, as a generalized quantum probability, naturally forms a quantum basic probability distribution and can be used to represent quantum measurement outcomes for quantum basic probability level decision-making. Furthermore, the quantum plausibility function in QET also can be naturally used to express the quantum measurement outcomes for quantum plausibility level decision-making.
3. These findings enrich the physical understanding of quantum amplitude encoding and quantum measurement outcomes.
Overall, this work is a generalized Bayesian inference method and is very meaningful to offer broad application prospects for representing and processing uncertain knowledge in pattern classification.

Weaknesses
1. As the authors mentioned in the paper. The current version of AQC-DRC is unable to process complex-valued input data, thereby constraining its applicability.
2. Why the quantum plausibility function in QET can be used to represent quantum measurement outcomes for quantum basic probability level decision-making, and can be naturally used to express the quantum measurement outcomes for quantum plausibility level decision-making?
3. The section of Preliminaries is too long. The content should be concise.

---

> ### Author Rebuttal · Authors · 2025-07-30
>
> Weakness:
>
> W1: Why the quantum plausibility function in QET can be used to represent quantum measurement outcomes for quantum basic probability level decision-making, and can be naturally used to express the quantum measurement outcomes for quantum plausibility level decision-making?
>
> A1: We sincerely appreciate your constructive feedback.
> In our concluding section, we mentioned “The quantum basic probability in QET, forming the quantum basic probability distribution, can directly express quantum measurement outcomes for basic probability level decision-making, while the quantum plausibility function in QET can also naturally represent the quantum measurement outcomes for plausibility-level decision-making.” This section presents two conclusions:
> 1)	The quantum basic probability in QET, forming the quantum basic probability distribution, can directly express quantum measurement outcomes for basic probability level decision-making.
> According to Definition 16, we obtain the QBP expression for quantum measurement outcomes after a rigorous mathematical proof. In other words, quantum measurement outcomes generated by $U_M^{QBP}$ can be simply computed to obtain QBP for basic probability level decision-making.
> 2)	The quantum plausibility function in QET can also naturally represent the quantum measurement outcomes for plausibility-level decision-making.
> Similarly, according to Definition 18, we obtain the QPl expression for quantum measurement outcomes after rigorous mathematical proof. In other words, quantum measurement outcomes generated by $U_M^{QPl}$ can be simply calculated to obtain QPl. Based on these two reasons, we arrive at these two conclusions mentioned in the conclusion section.
>
> W2: The section of Preliminaries is too long. The content should be concise.
>
> A2: Thank you for your suggestion. We agree that Preliminaries should be more concise. We have made changes to this section that will be presented in the final draft.
>
> Question:
>
> Q1: Why the quantum basic probability amplitude function in QET can be naturally utilized to express the quantum amplitude encoding?
>
> A1: We sincerely appreciate your constructive feedback.
> The quantum basic probability amplitude (QBPA) function is defined in Definition 9. According to this definition, the QBPA must satisfy the conditions outlined in Eq. (12), one of which is: $\sum_{\ket{\psi_j}\in \ket{\Phi}}|\mathbf{Q_M}(\ket{\psi_j})|^2=1$. In quantum computing, a superposition state must satisfy the condition that the sum of the squares of each mods of amplitude equals one, consistent with the conditions in Eq. (12). Therefore, the quantum basic probability amplitude function in QET can be naturally utilized to express the quantum amplitude encoding.
>
> Q2: What’s different between the quantum basic probability level decision-making and quantum plausibility level decision-making?
>
> A2: Thank you for your questions and suggestions.
> 1) The decision-making level of quantum basic probability can describe uncertain, partially unknown, and completely unknown quantum probabilities. For a three-class frame of discernment {$\phi_1, \phi_2, \phi_3$}, using 3 qubits as an example: The measurement result $\ket{001}$ represents the quantum probability of "supporting class $\phi_1$"; The measurement result $\ket{011}$ represents the quantum probability of an "uncertain state between class $\phi_1$ and class $\phi_2$"; The measurement result $\ket{111}$ represents the quantum probability of being "completely unknown (uncertain state among class $\phi_1$, class $\phi_2$ and class $\phi_3$)". The decision-making level of quantum plausibility can provide an upper bound on quantum probability, representing an optimistic estimate. This level can be used mainly in scenarios with high security requirements, such as medical diagnostics or military intelligence analysis. For example, event $\phi_1$ represents an infectious disease. The quantum basic probability (QBP) of a person contracting the disease is M$(\phi_1) = 0.3$ that is lower than 0.5, whereas the quantum plausibility (QPl) is QPl$(\phi_1) = 0.95$. Although the person cannot be directly identified as having the disease, the possibility that the person has the disease should still not be ignored for safety reasons.
> 2) In addition, from the computational complexity point of view, the computational complexity of AQC-DRC_QPl is $O(kn^2)$, while the computational complexity of AQC-DRC_QBP is $O(kn+N)$. Therefore, if the usage scenario of the algorithm does not require significant bias towards one of the two levels, one option is to select an algorithm with lower computational complexity, based on the relationship between the size of FOD $n$ and the number of the focal elements $N$.
> Your suggestion made us realize that our explanation of this part was not clear enough, and we have made changes to this part that will be presented in the final draft.
>
> Q3: Figure A1 is a main flowchart of the adaptive quantum circuit for Dempster’s rule of combination. It should be placed in the main text.
>
> A3: We sincerely appreciate your constructive feedback.
> We agree that Figure A1, which illustrates the main flowchart of the adaptive quantum circuit for Dempster’s rule of combination, plays an important role. In the revised manuscript, if space permits, we will move this figure from the appendix to the main text to improve clarity and help readers better understand the overall workflow and structure of the algorithm.

---

> > ### Comment · Reviewer_rA57 · 2025-08-05
> >
> > Thanks for the authors' thorough and detailed rebuttal. I appreciate the clear explanation of how the quantum basic probability amplitude function in QET naturally lends itself to quantum amplitude encoding. The clarification on the role of the quantum plausibility function in capturing measurement outcomes across both the basic probability and plausibility levels effectively addresses my concerns regarding the interpretation of decision-making processes in QET. The distinction between quantum basic probability level and plausibility level decision-making is clearly articulated and theoretically grounded. These clarifications substantially strengthen the conceptual foundation of their work. Based on these improvements, I am confident that the concerns raised in my initial review have been adequately addressed.

---

> > > ### Author Response · Authors · 2025-08-05
> > > **Response to Reviewer Comments**
> > >
> > > Thank you very much for your valuable suggestions. Your review comments have improved the quality of our manuscript, and we are very grateful for them!

---

### Official Review · Reviewer_FwDi · 2025-06-24

**Clarity:** 2
**Significance:** 2
**Originality:** 2
**Rating:** 4
**Confidence:** 4

**Summary:**

The manuscript proposes a so-called quantum version of the DS evidence framework, and suggests implementing the model's computation using adaptive quantum circuits. However, there some fundamental misuses in the basic formalization of the manuscript.

**Questions:**

Essentially, the probabilistic structure corresponding to quantum mechanics is a non-classical probabilistic structure that relaxes the closure property of sigma algebra with respect to countable unions in Kolmogorov's probability axioms. Does the authors of the paper know that this structure is not isomorphic to the mathematical structure underlying the DS theory, and therefore cannot be directly expressed in a quantum formalism?

**Ethical Concerns:**

["NO or VERY MINOR ethics concerns only"]

**Final Justification:**

The research motivation and significance are not particularly impressive.

**Limitations:**

Essentially, the probabilistic structure corresponding to quantum mechanics is a non-classical probabilistic structure that relaxes the closure property of sigma algebra with respect to countable unions in Kolmogorov's probability axioms. Does the authors of the paper know that this structure is not isomorphic to the mathematical structure underlying the DS theory, and therefore cannot be directly expressed in a quantum formalism?

**Quality:**

3

**Strengths And Weaknesses:**

In the main mathematical formalization of the manuscript, there exist misuses (or at least a misleading uses). For example, under the specification of Dirac's bra-ket notation, |ϕ1ϕ2> refers to the simultaneous occurrence of events ϕ1 and ϕ2 on the (two) subsystems of the composite system, respectively. However, the |ϕ1ϕ2> defined in the manuscript (in accordance with the convention of Dempster-Shafer theory) refers to the occurrence of either ϕ1 or ϕ2 on the same system. Overall, although the formalization in the manuscript employs the fundamental formalization of quantum mechanics (Dirac's bra-ket notation), it is contrary to the intrinsic interpretation of the formalization of quantum mechanics in terms of substantive meaning.

Because this misuse was fundamental, the subsequent formalizations (Definitions 8, 9 and 10, etc.) in the manuscript also deviated from the essential meaning of quantum mechanics. Therefore, the entire formalization of the manuscript and its algorithm model should not be regarded as being quantum in nature. It is merely a heuristic method that shares some formal characteristics with quantum mechanics, but its substantive meaning contradicts the fundamental interpretation of quantum mechanics.

---

> ### Author Rebuttal · Authors · 2025-07-30
>
> Weakness:
>
> W1: In the main mathematical formalization of the manuscript, there exist misuses (or at least a misleading uses). For example, under the specification of Dirac's bra-ket notation, $\ket{\phi_1\phi_2}$ refers to the simultaneous occurrence of events $\phi_1$ and $\phi_2$ on the (two) subsystems of the composite system, respectively. However, the $\ket{\phi_1\phi_2}$ defined in the manuscript (in accordance with the convention of Dempster-Shafer theory) refers to the occurrence of either $\phi_1$ or $\phi_2$ on the same system. Overall, although the formalization in the manuscript employs the fundamental formalization of quantum mechanics (Dirac's bra-ket notation), it is contrary to the intrinsic interpretation of the formalization of quantum mechanics in terms of substantive meaning.
> Because this misuse was fundamental, the subsequent formalizations (Definitions 8, 9 and 10, etc.) in the manuscript also deviated from the essential meaning of quantum mechanics. Therefore, the entire formalization of the manuscript and its algorithm model should not be regarded as being quantum in nature. It is merely a heuristic method that shares some formal characteristics with quantum mechanics, but its substantive meaning contradicts the fundamental interpretation of quantum mechanics.
>
> A1: Thank you very much for your constructive and rigorous comments. We would like to respond as follows:
> 1) Inspired by quantum probability theory, which generalizes classical probability theory within the mathematical framework of quantum mechanics, we proposed a quantum evidence theory that integrates Dempster-Shafer (DS) evidence theory with quantum probability. The aim is to address uncertainty and ambiguity in complex decision tasks.
> 2) In doing so, we discussed the mathematical equivalence of fundamental concepts in quantum evidence theory within Hilbert space, to further explore the intrinsic relationship between quantum evidence theory and the mathematical framework of quantum mechanics. We sincerely apologize for any confusion this may have led to.
> 3) Just as classical probability evolved into quantum probability, and Euclidean geometry extended into Riemannian geometry, we are inspired to propose quantum evidence theory as a natural extension for more complex application scenarios. On one hand, it has successfully modeled interference effects observed in classification decision task. On the other hand, the notions of quantum basic probability amplitude and quantum basic probability give new physical meaning to quantum amplitude encoding and quantum measurement from an information-theoretic perspective.
> 4) In the revised manuscript, we will clearly state that the expression $\phi_1\phi_2$ denotes a composite quantum proposition, and that the corresponding ket notation $\ket{\phi_1\phi_2}$ is a mathematical representation in Hilbert space, equivalent to how composite quantum proposition are encoded.
> In summary, we appreciate your critical insight and will make sure to clarify these distinctions in the revision to avoid any misinterpretation of the model’s intent and theoretical nature.
>
> Question:
>
> Q1: Essentially, the probabilistic structure corresponding to quantum mechanics is a non-classical probabilistic structure that relaxes the closure property of sigma algebra with respect to countable unions in Kolmogorov's probability axioms. Does the authors of the paper know that this structure is not isomorphic to the mathematical structure underlying the DS theory, and therefore cannot be directly expressed in a quantum formalism?
>
> A1: Thank you for your deep and insightful question.
> 1) As we answered for W1, inspired by quantum probability theory, which generalizes classical probability theory within the mathematical framework of quantum mechanics, we proposed a quantum evidence theory that integrates Dempster-Shafer (DS) evidence theory with quantum probability. The aim is to address uncertainty and ambiguity in complex decision tasks. In addition, we discussed the mathematical equivalence of fundamental concepts in quantum evidence theory within Hilbert space, to further explore the intrinsic relationship between quantum evidence theory and the mathematical framework of quantum mechanics.
> 2) For example, $\phi_1\phi_2$ denotes a composite quantum proposition, and that the corresponding ket notation $\ket{\phi_1\phi_2}$ is a mathematical representation in Hilbert space, equivalent to how composite quantum proposition are encoded. Take a two-class frame of discernment {$\phi_1, \phi_2$}, using 2 qubits as an example: The measurement result $\ket{11}$, corresponding to composite quantum proposition $\phi_1\phi_2$ mathematical representation $\ket{\phi_1\phi_2}$ in Hilbert space, represents the quantum probability for composite quantum proposition $\phi_1\phi_2$. The measurement result $\ket{11}$ represents the quantum probability of being "completely unknown (uncertain state between class $\phi_1$ and class $\phi_2$)".
> We appreciate your critical insight. And, we will modify and clarify this positioning more explicitly in the revised manuscript to prevent any misunderstanding regarding the nature of our formalism.

---

> > ### Comment · Reviewer_FwDi · 2025-08-08
> > **Official Comment by Reviewer FwDi**
> >
> > Thank you for your reply. Although you explained in your reply the difference between the interpretation of your formalism and the standard understanding of the state vector formalism of quantum mechanics, there are still some open technical issues. For example, in your reply you suggest using the measurement result |11> to represent the quantum probability of being "completely unknown (uncertain state between class $\phi_1$ and class $\phi_2$)".  Therefore, I understand that the target task you are modeling is essentially a one-hot classification problem (which matches the standard problem setting of DS-theory). The point this raises is why would you use an exponentially large state space to model an intrinsic linear problem space? Specifically, for a three-class classification task, your task space has only three possible states:  (1, 0, 0), (0, 1, 0) and (0, 0, 1). So, is it necessary to use an exponentially large state space (here it is (0,0,0), (0,0,1), (0,1,0),...,(1,1,1)) to model it?
> > In short, I think your model is just a heuristic that shares some computational rules with the quantum mechanical formalism. But their respective physical interpretations are clearly different. At present, I also fail to see the necessity of using the quantum mechanical formalism in the context of your task.  Hence I maintain my original assessment. However, this is my last comment on this paper. If other reviewers think this paper should be accepted, I will not argue further.

---

> > > ### Author Response · Authors · 2025-08-08
> > > **Response to Reviewer Comments**
> > >
> > > Q1: Thank you for your reply. Although you explained in your reply the difference between the interpretation of your formalism and the standard understanding of the state vector formalism of quantum mechanics, there are still some open technical issues. For example, in your reply you suggest using the measurement result $\ket{11}$ to represent the quantum probability of being "completely unknown (uncertain state between class $\phi_1$ and class $\phi_2$)". The point this raises is why would you use an exponentially large state space to model an intrinsic linear problem space?
> > >
> > > A1: Thank you for your question. In fact, in Dempster-Shafer evidence theory (DSET), when the size of the frame of discernment is $n$, the number of propositions involved in the combination is $2^n$, which is not a linear-level problem as you mentioned, but an exponential-level problem. This is the basic definition of DSET, and it is clearly stated in Definition 1 in the “Preliminaries” section of the manuscript. Therefore, your statement “use an exponentially large state space to model an intrinsic linear problem space” is a misunderstanding. This section does not contain the open technical issues you mentioned. We actually used an exponentially large state space to model an exponentially large state space.
> > >
> > > Q2: Specifically, for a three-class classification task, your task space has only three possible states: (1, 0, 0), (0, 1, 0) and (0, 0, 1). So, is it necessary to use an exponentially large state space (here it is (0,0,0), (0,0,1), (0,1,0),...,(1,1,1)) to model it?
> > >
> > > A2: Thank you for your question. In the three-classification problem you mentioned, the possible states are not limited to the 3 you mentioned, but include (0,0,0), (0,0,1), (0,1,0), (0,1,1), (1,0,0), (1,0,1), (1,1,0), and (1,1,1), corresponding to the propositions {$\emptyset$}, {$\phi_1$}, {$\phi_2$}, {$\phi_1, \phi_2$}, {$\phi_3$}, {$\phi_1, \phi_3$}, {$\phi_2, \phi_3$}, and {$\phi_1, \phi_2, \phi_3$} in DSET, totaling $2^3$.  The single subset {$\phi_1$} represents "supporting class $\phi_1$"; The composite subset {$\phi_1, \phi_2$} represents  "uncertain state between class $\phi_1$ and class $\phi_2$"; The full frame of discernment {$\phi_1, \phi_2, \phi_3$} represents "completely unknown (uncertain state among class $\phi_1$, class $\phi_2$ and class $\phi_3$)". In quantum computing, a three qubits system also has $2^3$ basis states, which is exactly the same as the number of propositions. Therefore, the statement “your task space has only three possible states” is a misunderstanding, and it’s necessary to use an exponentially large state space to model it.
> > >
> > > Q3: In short, I think your model is just a heuristic that shares some computational rules with the quantum mechanical formalism. But their respective physical interpretations are clearly different. At present, I also fail to see the necessity of using the quantum mechanical formalism in the context of your task.
> > >
> > > A3: Thank you for your comment. The statement “I think your model is just a heuristic that shares some computational rules with the quantum mechanical formalism” is a misunderstanding. Our algorithm is not a heuristic algorithm, but rather a quantum circuit of a classical algorithm (DRC) that utilizes quantum parallelism to reduce complexity exponentially, which is the main contribution of our work. Assume that there are $n$ elements in the frame of discernment and $k$ pieces of evidence, with a total of $N$ focal elements. The computational complexity of the DRC and DRC+QPl for decision-making is both $O(kN2^{2n})$. The computational complexity of the proposed adaptive quantum circuit for Dempster's rule of combination using QPl for decision-making is $O(kn^2)$. Given that the computational complexity of DRC is $O(kN2^n)$ and that of the proposed AQC-DRC_QPl is $O(kn^2)$, it is clear that our algorithm's computational complexity has decreased exponentially. Therefore, the use of quantum computing in the combination rule of DSET is very meaningful, rather than unnecessary.

---

> > > > ### Comment · Reviewer_FwDi · 2025-08-09
> > > > **Official Comment by Reviewer FwDi**
> > > >
> > > > Thank you for your reply and clarification. Based on your new response, I now realize that I had some misunderstandings on your previous reply. However, my main concern regarding the formalization of your paper still remains. In short, in the DS ( or QDS defined by you), all subsets of Ω (or {|\Phi>) are supposed to be jointly measurable (as stated in the formula (4) and (12) in your paper). Joint measurability is actually a fundamental characteristic of classical (Kolmogorovian) probability. However, quantum mechanics is contextual. That is, in general, quantum random variables are not jointly measurable. To be specific, let Ω={h1,h2.h3,h4}. Then, according to QM, different (projection) measurement contexts (i.e., measurable subspaces) can be defined. E.g., {(1,0,0,0), (0,1,0,0), (0,0,1,0), (0,0,0,1)} and  {(1,0,0,1)/c, (1,0,0,-1)/c, (0,1,1,0)/c, (0,1,-1,0)/c} are two different quantum measurement contexts, which satisfy, e.g., the probability normalization condition, respectively. If you use quantum formalism to model DS theory, in principle, you need map every measurement vectors in the above two measurement contexts to some subset of Ω (or |\Phi>), respectively. Furthermore, the subsets mapped to each measurement context should be jointly measurable (e.g., they should satisfy the probability normalization condition, respectively). However, the inconsistency of your formalization lies in the fact that DS requires that all 2^4 subsets of Ω (or |\Phi>) are jointly measurable (e.g, all 2^4 subsets of Ω (or |\Phi>) must satisfy the probability normalization condition as required by the formula (4) and (12) in your paper). Therefore, according to the definition of formula (12), your model is fundamentally still a classical probability model that is jointly measurable, rather than a quantum probability model w.r.t measurement contexts. Calling this kind of definition "quantum" would cause considerable confusion in thinking.

---

> ### Author Response · Authors · 2025-08-09
> **Response to Reviewer Comments**
>
> Q1: Thank you for your reply and clarification. Based on your new response, I now realize that I had some misunderstandings on your previous reply. However, my main concern regarding the formalization of your paper still remains. In short, in the DS ( or QDS defined by you), all subsets of Ω (or {|\Phi>) are supposed to be jointly measurable (as stated in the formula (4) and (12) in your paper). Joint measurability is actually a fundamental characteristic of classical (Kolmogorovian) probability. However, quantum mechanics is contextual. That is, in general, quantum random variables are not jointly measurable. To be specific, let Ω={h1,h2.h3,h4}. Then, according to QM, different (projection) measurement contexts (i.e., measurable subspaces) can be defined. E.g., {(1,0,0,0), (0,1,0,0), (0,0,1,0), (0,0,0,1)} and {(1,0,0,1)/c, (1,0,0,-1)/c, (0,1,1,0)/c, (0,1,-1,0)/c} are two different quantum measurement contexts, which satisfy, e.g., the probability normalization condition, respectively. If you use quantum formalism to model DS theory, in principle, you need map every measurement vectors in the above two measurement contexts to some subset of Ω (or |\Phi>), respectively. Furthermore, the subsets mapped to each measurement context should be jointly measurable (e.g., they should satisfy the probability normalization condition, respectively). However, the inconsistency of your formalization lies in the fact that DS requires that all 2^4 subsets of Ω (or |\Phi>) are jointly measurable (e.g, all 2^4 subsets of Ω (or |\Phi>) must satisfy the probability normalization condition as required by the formula (4) and (12) in your paper). Therefore, according to the definition of formula (12), your model is fundamentally still a classical probability model that is jointly measurable, rather than a quantum probability model w.r.t measurement contexts. Calling this kind of definition "quantum" would cause considerable confusion in thinking.
>
> A1：Thank you for your comment. The statement that “calling this kind of definition ‘quantum’ would cause considerable confusion in thinking” seems to be a misunderstanding. Quantum evidence theory is founded upon both quantum probability theory [1][2] and Dempster-Shafer evidence theory [3][4]. Given that our work is interdisciplinary, we intentionally use the term “quantum” to highlight the quantum properties involved, consistent with similar usages as [5][6]. Therefore, we have named it the “quantum basic probability amplitude function.”
>
> [1] Stanley P Gudder. Quantum probability. Academic Press, 2014.
>
> [2] Miklós Rédei and Stephen Jeffrey Summers. Quantum probability theory. Studies in History and Philosophy of Science Part B: Studies in History and Philosophy of Modern Physics, 38(2):390–417, 2007.
>
> [3] AP DEMPSTER. Upper and lower probabilities induced by a multivalued mapping. Annals of Mathematical Statistics, 38:325–339, 1967.
>
> [4] Glenn Shafer. A mathematical theory of evidence, 1976.
>
> [5] Eren Utku. An introduction to quantum probability amplitude modulation (qpam) from a compositional perspective. In ADVANCES IN QUANTUM COMPUTER MUSIC, pages 117–132. World Scientific, 2025.
>
> [6] Shahram Payandeh. Applications of quantum probability amplitude in decision support systems. Applied Computational Intelligence and Soft Computing, 2023(1):5532174, 2023.

---

### Official Review · Reviewer_XsYs · 2025-07-02

**Clarity:** 3
**Significance:** 3
**Originality:** 3
**Rating:** 5
**Confidence:** 1

**Summary:**

The authors propose an Adaptive Quantum Circuit for Dempsters Rule of Combination (AQC-DRC) calculations in Dempster-Shafer (DS) theory of evidence which combines two basic probability assignments (BPA).  After defining basic DS concepts such as evidence, plausibility, BBA, etc. in quantum terms, the lay out their Adaptive Quantum Circuit which consists of a quantum amplitude encoding for BPA, a construction of the adaptive quantum circuit for DRC, and a measurement in the adaptive quantum circuit for subsequent decision-making.  They implement methods for basic probability and plausibility with the AQC-DRC.  They evaluate their quantum system on three classification datasets comparing their approach with the classical DS classification and show that the quantum methods gradually approach the classical results as the number of measurement shots increases while achieving an exponential speedup over classical methods.

**Questions:**

Have you thought about mapping conditional beliefs and plausibilities into the quantum framework, i.e., Bl(A|B) and Pl(A|B)?  These are particularly useful as lower and upper bounds for Bayesian conditionals P(A|B).  Also, can you please clarify how you arrived at the various quantum complexity results?

**Ethical Concerns:**

["NO or VERY MINOR ethics concerns only"]

**Final Justification:**

I have no more updates based on the authors' feedback.

**Limitations:**

I would move the limitations section to the regular text

**Quality:**

3

**Strengths And Weaknesses:**

Strengths:

The mapping of classical DRC into a quantum framework and then showing the viability and exponential speedup without significant loss of accuracy is important for the future when quantum computers will have evolved to a point where the are available for performing the types of classifications tasks the authors discuss.

Weakness:

The DRC is know to be problematic and I wonder if the authors could also implement the Fagin-Halpern conditional which provides a better method for information fusion without the problems of the DCR.  It would have been also good to discuss any potential near-term applications of their AQC-DRC.

---

> ### Author Rebuttal · Authors · 2025-07-30
>
> Weakness:
>
> W1: The (Dempster rule of combination) DRC is know to be problematic and I wonder if the authors could also implement the Fagin-Halpern conditional which provides a better method for information fusion without the problems of the DCR.
>
> A1: We sincerely appreciate your constructive feedback.
> Due to space constraints, the implementation of the quantum Fagin-Halpern conditional method will be considered in our near future work.
>
> W2: It would have been also good to discuss any potential near-term applications of their AQC-DRC.
>
> A2: We sincerely appreciate your constructive feedback.
> We agree that discussing near-term applications of the AQC-DRC would add value. In the short term, AQC-DRC could be applied to areas such as:
> 1) quantum-enhanced sensor fusion, where uncertain and conflicting measurements from multiple sources must be integrated efficiently;
> 2) decision-making under uncertainty in domains like autonomous systems, medical diagnosis, and threat assessment.
> We plan to discuss them in the future work and explore them in future work.
>
> Questions:
>
> Q1: Have you thought about mapping conditional beliefs and plausibilities into the quantum framework, i.e., $Bl(A|B)$ and $Pl(A|B)$?
>
> A1: Thank you for the insightful question.
> Yes, the idea of mapping conditional beliefs and plausibilities $Bl(A|B)$ and $Pl(A|B)$ into the quantum framework is both important and promising. In future work, we plan to explore quantum analogues of conditional belief and plausibility to reflect updated knowledge given new evidence. This would further enhance the expressive power and interpretability of quantum-based reasoning systems.
>
> Q2: Also, can you please clarify how you arrived at the various quantum complexity results?
>
> A2: We sincerely appreciate your constructive feedback.
> All the algorithms in Table 2 of the manuscript can be divided into two parts: the combination process, which involves obtaining the complete combined evidence or QPl; and the decision-making process, which involves using the obtained evidence or QPl to reach a decision. Assume that there are $n$ elements in the frame of discernment (FOD) and $k$ pieces of evidence, with a total of $N$ focal elements.
> 1) DRC: When combining two pieces of evidence, the computational complexity of computing any of the focal elements is $O(2^{2n})$. Since there are $N$ focal elements in total, the computational complexity of getting the complete combination result is $O(N2^{2n})$. The total number of combinations is $k-1$, so the total complexity of combination process is $O(kN2^{2n})$. After the combination process, a decision is made from n elements in the FOD. The computational complexity of the decision-making process is $O(n)$. Thus, the total computational complexity of DRC is $O(kN2^{2n})$. ($O(kN2^{2n}+n)=O(kN2^{2n})$)
> 2) DRC+QPl: The computational complexity required to compute the QPl of a focal element from QBP in the classical process is $O(N)$. In the experimental process we made the decision to use the QPl of elements in FOD, so the computational complexity to compute QPl is $O(nN)$. Since the computational complexity of the DRC is $O(kN2^{2n})$, the total computational complexity of combination process is $O(kN2^{2n})$. ($O(kN2^{2n}+nN)=O(kN2^{2n})$, as $nN<kN2^{2n}$ when $n≥2$). The computational complexity of decision-making process is $O(n)$. Thus, the total computational complexity of DRC+QPl is $O(kN2^{2n})$.
> 3) QC-DRC_QBP: The computational complexity of the quantum line part we use the depth of the quantum circuit to denote. The QC-DRC_QBP can only combine two pieces of evidence at a time. For the combination of two evidences the number of Toffoli gates used is $n$, so the computational complexity of the quantum circuit part is $O(n)$. There are $N$ measurements obtained from quantum circuit, each of which requires a normalization process of complexity $O(1)$, so the computational complexity of the normalization part is $O(N)$. The computational complexity of two evidence combinations is $O(n+N)$. In total, $k-1$ combinations have to be performed, so the computational complexity of combination process is $O(kn+kN)$. The computational complexity of decision-making process is $O(n)$. Thus, the total computational complexity of QC-DRC_QBP is $O(kn+kN)$.
> 4) QC-DRC_QBP+QPl: The computational complexity of QC-DRC_QBP is $O(kn+kN)$, and the computational complexity of the classical process of generating QPl from QBP is $O(nN)$. Therefore, the total computational complexity of combination process is $O(kn+kN+nN)$. The computational complexity of decision-making process is $O(n)$. Thus, the total computational complexity of QC-DRC_QBP+QPl is $O(kn+kN+nN)$.
> 5) AQC-DRC_QBP: The AQC-DRC_QBP can combine more than one piece of evidence at a time and the number required CNOT gate is $(k-1)n$. Thus, the complexity of the quantum circuit is $O(kn)$. The complexity of the normalization part is consistent with QC-DRC_QBP and is $O(N)$. Thus the total computational complexity of combination process is $O(kn+N)$. The computational complexity of decision-making process is $O(n)$. Thus, the total computational complexity of AQC-DRC_QBP is $O(kn+N)$.
> 6) AQC-DRC_QBP+QPl: The computational complexity of AQC-DRC_QBP is $O(kn+N)$, and the computational complexity of the classical process of generating QPl from QBP is $O(nN)$. Therefore, the total computational complexity of combination process is $O(kn+nN)$. ($O(kn+N+nN)=O(kn+(n+1)N)=O(kn+nN)$). The computational complexity of decision-making process is $O(n)$. Thus, the total computational complexity of AQC-DRC_QBP+QPl is $O(kn+nN)$.
> 7) AQC-DRC_QPl: The algorithm involves measuring $n$ qubits individually, so a total of $n$ quantum circuits are needed to complete it. The computational complexity of each of these quantum circuits is consistent with that of the quantum circuit of AQC-DRC_QBP, which is $O(kn)$, and thus its total computational complexity of all quantum circuits is $O(kn^2)$. The computational complexity of decision-making process is $O(n)$. Thus, the total computational complexity of AQC-DRC_QPl is $O(kn^2)$.
>
> Limitations:
>
> L1: I would move the limitations section to the regular text.
>
> A1: Thank you for your suggestion. We agree that limitations should be included in the main text. We have made changes that will be presented in the final draft.

---

> > ### Comment · Reviewer_XsYs · 2025-08-05
> > **Complexity results**
> >
> > Thank you for providing the details on the complexity results, please add those to the paper, these are valuable additions.

---

> > > ### Author Response · Authors · 2025-08-05
> > > **Response to Reviewer Comments**
> > >
> > > Thank you very much for your valuable suggestions. If space permits, we will include those contents in the main text. If space is insufficient, we will include those contents in the appendix. Your review comments have improved the quality of our manuscript, and we are very grateful for them!

---

### Official Review · Reviewer_9Y4M · 2025-07-06

**Clarity:** 4
**Significance:** 4
**Originality:** 4
**Rating:** 6
**Confidence:** 5

**Summary:**

To address efficient classification under uncertain environments, the authors proposed an adaptive quantum circuit for Dempster’s rule of combination (AQC-DRC) to support quantum basic probability and plausibility level decision-making within the framework of quantum evidence theory (QET).

**Questions:**

1. What are the QC-DRC_QBP + QPl and AQC-DRC_QBP + QPl? The methods of QC-DRC_QBP + QPl and AQC-DRC_QBP + QPl should be explained in the text.
2. What are the purpose of two decision-making levels of quantum basic probability and quantum plausibility?
3. In what situations are quantum basic probability and quantum plausibility used?

**Ethical Concerns:**

["NO or VERY MINOR ethics concerns only"]

**Limitations:**

Yes

**Paper Formatting Concerns:**

There are no concerns regarding the paper formatting.

**Quality:**

4

**Strengths And Weaknesses:**

Strengths：-- The AQC-DRC, as a generalized quantum Bayesian inference method, enabled deterministic computation of DRC, ensuring that quantum fusion outcomes in uncertain pattern classification are fully consistent with those of the classical method.
-- Furthermore, the proposed AQC-DRC achieved an exponential reduction in computational complexity, positioning it as a promising approach for real-time quantum multisource information fusion.
-- The architecture of the proposed AQC-DRC is conceptually straightforward and highly scalable, which facilitates its practical implementation.
-- The quantum basic probability amplitude function in QET can naturally express the quantum amplitude encoding. The quantum basic probability in QET, forming the quantum basic probability distribution, can directly express quantum measurement outcomes for basic probability level decision-making, while the quantum plausibility function in QET can also naturally represent the quantum measurement outcomes for plausibility-level decision-making.
-- These insights not only broaden the understanding of QET, but also provide a more intuitive physical interpretation of quantum amplitude encoding and quantum measurement outcomes.
Weaknesses：
Due to the limitations of quantum hardware, the proposed AQC-DRC is unable to handle high-dimensional data.

---

> ### Author Rebuttal · Authors · 2025-07-30
>
> Questions:
>
> Q1: What are the QC-DRC_QBP + QPl and AQC-DRC_QBP + QPl? The methods of QC-DRC_QBP + QPl and AQC-DRC_QBP + QPl should be explained in the text.
>
> A1: We sincerely appreciate your constructive feedback.
> The QC-DRC_QBP is a non-adaptive quantum circuit for Dempster's rule of combination, which is used to compute quantum basic probability (QBP) for quantum basic probability level decision-making. The QC-DRC_QBP + QPl is a method that first uses QC-DRC_QBP to obtain QBP and then uses the classical process to combine QBP into QPl. Similarly, AQC-DRC_QBP + QPl is a method that first uses AQC-DRC_QBP to obtain QBP and then uses the classical process to combine QBP into QPl.
> Your suggestion made us realize that our explanation of this part was not clear enough, and we have made changes to this part that will be presented in the final draft.
>
> Q2: What are the purpose of two decision-making levels of quantum basic probability and quantum plausibility?
>
> A2: We sincerely appreciate your constructive feedback.
> 1. The decision-making level of quantum basic probability can describe uncertain, partially unknown, and completely unknown quantum probabilities. For a three-class frame of discernment {$\phi_1, \phi_2, \phi_3$}, using 3 qubits as an example:
> (1) The measurement result $\ket{001}$ represents the quantum probability of "supporting class $\phi_1$";
> (2) The measurement result $\ket{011}$ represents the quantum probability of an "uncertain state between class $\phi_1$ and class $\phi_2$";
> (3) The measurement result $\ket{111}$ represents the quantum probability of being "completely unknown (uncertain state among class $\phi_1$, class $\phi_2$ and class $\phi_3$)". The decision-making level of quantum plausibility can provide an upper bound on quantum probability, representing an optimistic estimate. This level can be used mainly in scenarios with high security requirements, such as medical diagnostics or military intelligence analysis. For example, event $\phi_1$ represents an infectious disease. The quantum basic probability (QBP) of a person contracting the disease is M$(\phi_1) = 0.3$ that is lower than 0.5, whereas the quantum plausibility (QPl) is QPl$(\phi_1) = 0.95$. Although the person cannot be directly identified as having the disease, the possibility that the person has the disease should still not be ignored for safety reasons.
> 2. In addition, from the computational complexity point of view, the computational complexity of AQC-DRC_QPl is $O(kn^2)$, while the computational complexity of AQC-DRC_QBP is $O(kn+N)$. Therefore, if the usage scenario of the algorithm does not require significant bias towards one of the two levels, one option is to select an algorithm with lower computational complexity, based on the relationship between the size of FOD $n$ and the number of the focal elements $N$.
> Your suggestion made us realize that our explanation of this part was not clear enough, and we have made changes to this part that will be presented in the final draft.
>
> Q3: In what situations are quantum basic probability and quantum plausibility used?
>
> A3: We sincerely appreciate your constructive feedback.
> Quantum basic probability and quantum plausibility are used in decision-making and classification tasks under quantum frameworks, particularly when dealing with uncertainty, partial ignorance, or complete ignorance.
> 1. Quantum basic probability is applied when one needs to model and quantify uncertain, partially unknown, or completely unknown events. It provides a quantum basic probability, allowing more expressive representations than classical probabilities.
> 2. Quantum plausibility is used when an optimistic estimation or an upper bound of the quantum probability is required, such as in quantum decision fusion or quantum belief updating. It helps assess the maximum possible support for a quantum hypothesis or event, particularly useful in ambiguous scenarios.
> Together, quantum basic probability and quantum plausibility are essential in quantum information processing, and quantum decision theory.

---

> > ### Comment · Reviewer_9Y4M · 2025-08-05
> >
> > Thank you to the authors for their thoughtful and comprehensive rebuttal. All of my comments have been addressed appropriately, and I have no remaining concerns. I find the submission to be strong and maintain my original positive assessment.

---

> > > ### Author Response · Authors · 2025-08-05
> > > **Response to Reviewer Comments**
> > >
> > > Thank you very much for your valuable suggestions. Your review comments have improved the quality of our manuscript, and we are very grateful for them!

---

### Note · Authors · 2025-08-12

We sincerely thank the reviewers for their detailed feedback and constructive suggestions. We are pleased to see that most of the reviewers agree with the technique novelty, comprehensive evaluation, and effectiveness. We apologize for any omissions and have addressed all comments, hoping to resolve your concerns. Below is a brief summary of our responses.

For Reviewer 9Y4M:

1.	We provided specific explanations of the meanings of QC-DRC_QBP and AQC-DRC_QBP.

2.	We explained of two new decision-making levels and their use situations.

For Reviewer XsYs:

1.	We addressed the reviewer’s suggestions by planning to implement the quantum Fagin-Halpern conditional in future work and discussing potential near-term applications of AQC-DRC.

2.	We explained that the complexity results in Table 2. These contents will be added to the manuscript.

3.	We agreed with the reviewer and moved the limitations section into the main text, which will be presented in the final draft.

For Reviewer FwDi:

1.	We clarified that our work is inspired by quantum probability theory as a natural extension of classical evidence theory. In the revised manuscript, we will explain the meaning of the expression '$\phi_1 \phi_2$'.

2.	We clarified the misunderstanding surrounding the statement:

1\)	“use an exponentially large state space to model an intrinsic linear problem space”.

2\)	“your task space has only three possible states: (1, 0, 0), (0, 1, 0) and (0, 0, 1)”.

3\)	“your model is just a heuristic that shares some computational rules with the quantum mechanical formalism”.

4\)	“calling this kind of definition ‘quantum’ would cause considerable confusion in thinking”.

For Reviewer rA57:

1.	We have provided an explanation why QET can be naturally used to express the quantum measurement outcomes for two level decision-making. Also, we explained why the quantum basic probability amplitude function in QET can be naturally utilized to express the quantum amplitude encoding.

2.	We agree that Preliminaries should be more concise. We have made changes to this section that will be presented in the final draft.

3.	We agree that Figure A1 plays an important role. In the revised manuscript, if space permits, we will move this figure from the appendix to the main text.

Once again, we thank all reviewers for their invaluable suggestions and insights, which have significantly contributed to the improvement of our paper.

---

### Decision · Program_Chairs · 2025-09-17

**Decision:**

Accept (poster)

**Comment:**

This paper proposes an Adaptive Quantum Circuit for Dempster's Rule of Combination (AQC-DRC), a quantum algorithm designed to accelerate evidence fusion in pattern classification under uncertainty. The authors claim their method achieves an exponential reduction in computational complexity compared to classical Dempster-Shafer Theory (DST) while guaranteeing that the quantum fusion outcomes are deterministically identical to the classical ones. The work is framed within a proposed "Quantum Evidence Theory" (QET), which the authors argue provides new physical insights into quantum amplitude encoding and measurement. Three of the four reviewers ultimately supported acceptance, with two high-confidence reviewers rating it as "Strong Accept" and "Accept".

The paper's primary strengths, as highlighted by the reviewers, are its novelty and potential impact. It presents a concrete quantum algorithm for a well-established, computationally expensive classical method. Reviewer 9Y4M noted that the AQC-DRC "achieved an exponential reduction in computational complexity, positioning it as a promising approach for real-time quantum multisource information fusion." [9Y4M] Reviewer rA57 praised the fact that it "provides a deterministic computation of DRC, ensuring that quantum fusion outcomes in uncertain pattern classification are exactly aligned with those of the classical method." [rA57] The main weakness stems from a fundamental critique by Reviewer FwDi, who argued that the paper's formalism misuses quantum mechanical principles. This reviewer contended that the model is "fundamentally still a classical probability model that is jointly measurable, rather than a quantum probability model... Calling this kind of definition 'quantum' would cause considerable confusion in thinking." [FwDi] Other minor weaknesses include the near-term infeasibility on current quantum hardware and the use of DRC, which is known to be problematic in some scenarios.

The author rebuttal was comprehensive and effectively addressed most concerns. For reviewers 9Y4M, XsYs, and rA57, the authors provided detailed clarifications on terminology, the motivation for different decision-making levels, and the derivation of complexity results, leading all three to affirm their positive ratings. The most significant discussion was with Reviewer FwDi, which involved a multi-turn debate. The authors successfully refuted the reviewer's initial misunderstanding that they were modeling a linear problem with an exponential state space by clarifying that DST itself operates on an exponential power set. While the reviewer conceded this, they maintained a more philosophical objection regarding the formalism's "quantumness." In my assessment, while Reviewer FwDi's critique is technically nuanced and important for the quantum foundations community, the authors' primary claim—the design of a quantum circuit that correctly and efficiently implements DRC—stands. The strong support from two high-confidence experts in the area outweighs this more philosophical disagreement.

Given the clear technical contribution, the demonstrated theoretical speedup, and the strong support from a majority of the reviewers, I recommend acceptance. The paper is technically solid and presents a novel application of quantum computing to a known ML-related problem. The debate surrounding the foundational interpretation of the formalism, while valid, does not invalidate the paper's core result concerning the quantum circuit implementation. The work is a good fit for the conference, contributing to the growing intersection of quantum computing and machine learning.